



# Characterizing and reducing equifinality by constraining a distributed catchment model with regional signatures, local observations, and process understanding

Christa Kelleher[1,2], Brian McGlynn [2], Thorsten Wagener [3,4]

5   [1]Department of Earth Sciences, Syracuse University, Syracuse, NY, 13244, USA
[2]Department of Earth and Ocean Sciences, Nicholas School of the Environment, Duke University, Durham, NC, 27706, USA
[3]Department of Civil Engineering, University of Bristol, Bristol, BS8 1TR, UK
[4]Cabot Institute, University of Bristol, Bristol, BS8 1TR, UK

10   *Correspondence to*: Christa Kelleher (ckellehe@syr.edu)

**Abstract.** Distributed catchment models are widely used tools for predicting hydrologic behaviour. While distributed models require many parameters to describe a system, they are expected to simulate behaviour that is more consistent with observed processes. However, obtaining a single set of acceptable parameters can be problematic, as parameter equifinality often results in several 'behavioural' sets that fit observations (typically streamflow). In this study, we investigate the extent to which equifinality impacts a typical distributed modelling application. We outline a hierarchical approach to reduce the number of behavioural sets based on regional, observation-driven, and expert knowledge-based constraints. For our application, we explore how each of these constraint classes reduced the number of 'behavioural' parameter sets and increased certainty in spatio-temporal simulations, simulating a well-studied headwater catchment, Stringer Creek, MT using the distributed hydrology-soil-vegetation model (DHSVM).   As a demonstrative exercise, we investigated model performance across 10,000 parameter sets. Constraints on regional signatures, the hydrograph, and two internal measurements of snow water equivalent time series further reduced the number of behavioural parameter sets, but still left a small number with similar goodness of fit. This subset was ultimately further reduced by incorporating pattern expectations of groundwater table depth across the catchment. Our results suggest that utilizing a hierarchical approach based on regional datasets, observations, and expert knowledge to identify behavioural parameter sets can reduce equifinality and bolster more careful application and simulation of spatio-temporal processes via distributed modelling at the catchment scale.

## 1 Introduction

The field of hydrology has been built upon the combination of field measurements and computational modelling to observe, understand, and predict hydrologic behaviour (Crawford and Linsley, 1966; Beven and Kirkby, 1976, 1979; Ponce and Shetty, 1995a, b).  Towards this end, distributed, physically based models were first developed as tools to represent

30   spatially discretized processes with physically meaningful parametric relationships (Freeze and Harlan, 1969; Beven and





Kirkby, 1979; Band et al., 1991, 1993; Wigmosta et al, 1994; Refsgaard and Storm, 1995; Bixio et al., 2002; Qu, 2004; Qu and Duffy, 2007; Camporese et al., 2010; Fatichi et al., 2016). Distributed models represent the characteristics of the catchment environment in space, given that this variability impacts how water is stored, partitioned, and released across the landscape (O'Loughlin, 1981; Beven, 1989; Kampf and Burges, 2007; Wagener et al., 2007; Nippgen et al., 2015). Many

physically-based, distributed models use similar equations to predict water movement, have comparably large data input requirements, and feature numerous model parameters (see Singh and Woolhiser (2002), Kampf and Burges (2007), or Paniconi et al. (2015) for a review of distributed models).

Despite their widespread application and advancement, one of the foremost challenges in distributed model application continues to be parameter estimation (Beven and Binley, 1992; Gupta et al., 1998; Wagener and Gupta, 2005;

Beven 2006; Gharari et al., 2014; Chen et al., 2017). Distributed model application requires the selection of (often) a single set or representative subset of parameters, which is especially challenging given that distributed models require a larger number of model parameters (~50-100 or more) and longer model run times than conceptual or lumped models. In particular, a large number of model parameters, corresponding to a large number of degrees of freedom, may lead to equifinality (Hornberger and Spear, 1980, 1981; Spear and Hornberger, 1980; Beven and Binley, 2014), a well-documented problem

with respect to distributed models (Beven 1989, 1993, 2001). Many studies have documented the presence of parameter equifinality in terms of parameter sensitivity, concluding that the potential for equifinality in distributed model applications is high in both time (Franks et al., 1997; van Werkhoven et al., 2008; Zhang et al., 2013, Kelleher et al., 2015; Ghasemizade et al., 2016; Guse et al., 2016) and space (Wagener et al., 2009; Herman et al., 2013; Moreau et al., 2013).

A great deal of research has been devoted to developing approaches that either seek to reduce or to explore

equifinality across multiple parameter sets, but these approaches are typically demonstrated for lumped or conceptual models (e.g., Keesman, 1990; van Straten and Keesman, 1991; Beven and Binley, 1992; Gupta et al, 1998). Currently, researchers approach distributed modelling and parameter set selection in a number of different ways, all of which have implications for reducing equifinality. Many studies avoid the topics of uncertainty or equifinality by assigning parameter values from measurements (e.g., Du et al., 2012), though the sheer number of parameters, heterogeneity of the catchment environment,

uncertainty in model structure and inputs, and problems translating measurements from the point to the grid scale can make this difficult (Grayson et al. 1992; Surfleet et al., 2011; Paniconi et al., 2015). Other approaches involve fixing some parameters, while letting others vary, often through manual calibration. The decision to fix versus vary some parameters often depends on the availability of field measurements, awareness of the catchment or model, and may involve sensitivity analysis to identify which parameters are most influential to simulations (Tang et al., 2007; Saltelli et al., 2008; Cuo et al.,

2011). A subset of distributed modelling studies have directly incorporated parameter uncertainty into final simulations, sampling across ranges of values obtained either from literature or measurements (e.g., Surfleet et al., 2010; Shields and Tague, 2012; Gharari et al., 2014; Silvestro et al., 2015). Very few studies have shown the implications of this equifinality during model calibration, an important consideration given that selection of a parameter set or sets will influence conclusions made for validation and scenario analysis.





Researchers must also determine which observations to incorporate and compare to model simulations (as well as which criteria to measure 'goodness of fit' by, and how to judge whether or not this 'goodness of fit' is good enough) in order to justify selection of a parameter set or sets (Bennett et al., 2013). While there are examples where different observations, including snow accumulation and melt (Thyer et al., 2004; Whitaker et al., 2003), soil moisture (Koren et al.,

2008; Graeff et al., 2012), and catchment chemistry (Birkel et al., 2014), have been incorporated into the calibration and parameter set selection process, many catchments lack measurements beyond streamflow, suggesting that other sources of information are needed (Yapo et al., 1998; Grayson et al., 2002; Paniconi et al., 2015). Finally, while examples exist where model simulations are compared to internal measurements of different hydrologic processes at a few points, there are fewer examples where researchers holistically evaluate the patterns of these simulated processes (Franks et al., 1998; Lamb et al.,

1998; Grayson et al., 2002; Wealands et al., 2005; Koch et al., 2016).

Given many of the outlined challenges discussed above, we need to improve our understanding of equifinality in distributed model applications and to better constrain this equifinality towards predictive use of distributed models. In this study, we demonstrate an approach to characterizing and reducing equifinality for distributed models. We apply this approach as a case study in the Stringer Creek headwater catchment, located in Tenderfoot Creek Experimental Forest in

central Montana, modelled with the widely applied Distributed Hydrology-Soil-Vegetation Model (DHSVM). We investigate how the paradigm for identifying 'behavioural' model simulations, defined as those that meet a certain criterion or multiple criteria for error with respect to observations, may impact the presence of equifinality in terms of parameter estimation and constraining simulations and predictions of different hydrologic processes. Within our proposed framework, we test many of the subjective choices a modeller must make during calibration with respect to impacts on parameter set

selection, parameter values, model performance, and model simulations. We explore model performance for a shorter period of time but for a large number of model runs. Our goal is to support distributed models utilization to their full potential and ensure that parameter set(s) used for prediction or scenario analysis match hydrologic observations and perceptions in both space and time. While our approach is demonstrated for a single catchment for a relatively short time period, it has broader implications for the use of alternative data sources in the parameter estimation process as well as the application of

distributed models to simulate internal catchment behaviour.

## 2.0 Case Study: Stringer Creek, Tenderfoot Creek Experimental Forest

### 2.1 Study Site

We present a case study applying this methodology to a headwater catchment (5.5 km$^2$) located in the Tenderfoot Creek Experimental Forest in central Montana. The case study focuses on Stringer Creek, though we simulate the entire

Tenderfoot Creek catchment (22.5 km$^2$). The Stringer Creek headwater catchment is located in Tenderfoot Creek Experimental forest in central Montana. Tenderfoot Creek experiences a continental climate with the majority of its 880mm of precipitation falling as snow from November through May (Farnes et al., 1995). Snowmelt, occurring in May or June, is



the primary driver of the hydrologic cycle. Bedrock under Stringer Creek includes biototite hornblende quartz monzonite overlaying a layer of shale in the upper parts of the catchment and flathead sandstone underlain by granite gneiss in the lower portions of the catchment (Reynolds, 1995). Lodgepole pine dominates the forest vegetation types, with shrubs and grasses occurring in riparian areas (Farnes et al., 1995; Mincemoyer and Birdsall, 2006). Stringer Creek, the most well

studied catchment in Tenderfoot Creek Experimental Forest, is a second-order catchment that drains an area of 5.5 km$^2$.

### 2.2 The Distributed Hydrology Soil Vegetation Model

The entire Tenderfoot Creek catchment was modelled using the Distributed Hydrology-Soil-Vegetation Model (Wigmosta et al., 1994; Wigmosta et al., 2002). DHSVM is a physically-based, spatially distributed catchment model typically used to simulate mountainous catchments at small to intermediate scales in the Pacific Northwest and the Mountain

West. The model includes a two-layer snow accumulation and melt model with a full energy balance and a Penman-Monteith approach for simulating evapotranspiration. Unsaturated soil water movement is simulated via Darcy's Law with hydraulic conductivity calculated via the Brooks-Corey equation. Saturated subsurface flow is routed cell-by-cell using either a kinematic or diffusion approximation. Streamflow is routed through a user-defined stream network via linear channel reservoirs. The model framework and forcing data used to simulate Tenderfoot Creek was previously employed and

described in Kelleher et al. (2015). Key details for spatial and meteorological forcing data are described below.

### 2.2.1 Spatial Forcing Data

Tenderfoot Creek was simulated within DHSVM at a resolution of 10m and a time step of 3 hours using spatially distributed information for topography, soil depth, soil type, and vegetation (Figure 1). Spatially distributed information for topography and vegetation height were obtained via airborne laser swath mapping (ALSM) at 1m resolution and resampled

to 10m resolution (Figure 1a, b). Soil depth was held at a constant one meter, as spatially distributed soil data is limited. However, >160 well and piezometer installations indicate that 1m is a reasonable average with limited variance (Jencso et al., 2009).

DHSVM distributes parameter values based on vegetation and soil 'types', with one vegetation and soil type assigned to each cell within the catchment. To minimize potential for equifinality, we employed a model framework to

distribute soil and vegetation parameters using as few different classes as possible while still representing functional differences that we expected to impact hydrology across the catchment. Vegetation was grouped into three classes based on vegetation height, as vegetation species are relatively homogenous across the catchment (Farnes et al., 1995; Mincemoyer and Birdsall, 2007) and height is important determinant of aerodynamic resistance, used within both the snow and evapotranspiration modules to estimate catchment response on a cell-by-cell basis (Wigmosta et al., 2002). Vegetation

classes included tall trees (>10m), trees (2m-10m), and undergrowth (<2m; Figure 1b). Vegetation parameters were distributed by vegetation type, with individual parameters partitioned between an understory (13 parameters) and an overstory (17 parameters). Cells with only undergrowth include only understory parameters; cells with a canopy (trees or tall





trees) have both an understory and an overstory. Parameter values associated with understory vegetation were varied in tandem across the catchment. Overstory parameters were varied in tandem for cells with both canopy and tall canopy vegetation, with the exception of vegetation height and overstory fractional coverage (aerial percentage of each cell covered by canopy vegetation) which we expected to differ between these two vegetation classes. Parameter values for vegetation

were constrained, when possible, based on the plant species at different heights (lodgepole pine when species information was available, more generally 'evergreen conifer' for 2 – 10m and 'grasses and shrubs' for <2m). All vegetation parameters are listed in Table 1, with ranges specified in Appendix A and sources for these ranges provided in Kelleher et al. (2015).

A single soil type was used to distribute soil parameters across the catchment, as there is limited small-scale information about soil properties across Stringer Creek. Parameter values for soil information were obtained, when possible,

from the CONUS soils database as well as from texture estimates from CONUS combined with the soil water characteristics hydraulic properties calculator following Saxton and Rawls (2006). Within DHSVM, a total of 15 parameters are used to describe soil characteristics for each soil type.

### 2.2.2 Meteorological Forcing Data

The model was executed at a time step of three hours to effectively simulate snowmelt while balancing

computational cost. Model forcing data includes air temperature, precipitation (which is partitioned between rain and snow using two temperature thresholds set by parameter values within the model framework), relative humidity, wind speed, and solar and longwave radiation (Figure 1c). All but solar and longwave radiation were measured continuously at two SNOTEL sites within Tenderfoot Creek Experimental Forest, with one site located at low elevation and another site located at high elevation. Air temperature was distributed across the basin using a lapse rate calculated at each model time step

between the two SNOTEL sites; all other meteorological information from SNOTEL sites was distributed using an inverse distance weighting approach contained within the model. Solar radiation data at the SNOTEL sites was sparse and discontinuous; instead, solar radiation measured at a FLUX tower was scaled to SNOTEL site locations using topographic position (Kelleher et al., 2015). Within the model, solar radiation is distributed across the catchment using monthly averaged shading maps for each model time step across a 24-hour period. Uniform shading maps were averaged for the months of

November, December and January, as spatially distributed shading maps produced simulations of climate and hydrology that exceeded allowable ranges for the model (Kelleher et al., 2015). As these months are primarily periods of accumulation and very little to no melt, we do not expect this to affect model results. Longwave radiation was calculated using the Stefan-Boltzmann equation (Dingman, 2002).

### 2.2.3 Model Initialization and Setup

For each parameter sample, the model was run for a six-month warm-up period from April 1, 2006 through September 30, 2006 and an analysis period of October 1, 2006 through October 1, 2008. The model was initialized using observations from the two SNOTEL stations. To test that the warm-up period was sufficiently long for the impact of initial





conditions on simulations to dissipate, we initialized nine parameter sets from varying initial conditions (results shown in Appendix A). Similar to other distributed model studies (e.g., Melsen et al., 2016), we found that model simulations diverged quickly, suggesting that six months should be sufficiently long for the influences of parameter values on model simulations and predictions to emerge. We focus on calibration to the 2008 water year, but demonstrate results for the 2007 water year to show the value of our approach across a longer period.

## 3.0 A Framework for Constraining Environmental Simulations and Predictions

### 3.1 Assessing Model Performance

There is a wealth of information and approaches researchers use to assess model performance and presence of equifinality, often by quantifying how well simulations match observations, aggregated catchment or regional data, and/or perceptions of system functioning (Bennett et al., 2013). Across the modelling and prediction in ungauged catchment literature, we have found that constraints on model-derived hydrologic behaviour generally fall into three general categories:

1. Regional signatures of similarity (e.g., Yadav et al., 2007; Bloeschl et al., 2013; Hrachowitz et al., 2014), typically applied to regionalize hydrologic models for streamflow prediction in ungauged catchments.
2. Objective functions or error metrics (e.g., van Werkhoven et al., 2008, Wagener et al., 2001; Gupta et al., 2008; Pfannerstill et al., 2014; Shafii and Tolson, 2015), which measure how well simulations match observations.
3. Measures of internal catchment behaviour, which describe how well simulations match either observations or perceptions of the spatio-temporal variability of hydrologic behaviour (e.g. Franks et al., 1998; Lamb et al., 1998; Grayson et al., 2002; Wealands et al., 2005; Kuras et al., 2011; Koch et al., 2016).

Whether within an uncertainty framework or applied to calibrating a single parameter set, most studies typically use one of the three types of constraints outlined above to obtain a 'best' parameter set; few have made use of more than one constraint type, though many studies have shown the value of multi-objective approaches (Yapo et al., 1998; Gupta et al., 1998; van Werkhoven et al., 2009). Recently, a number of studies have advanced approaches for assessing alternative information sources (i.e., model realism via expert knowledge) and constraining both parameter values and model simulations in pursuit of reducing equifinality (Seibert and McDonnell, 2002; Hrachowitz et al., 2014; Gharari et al., 2014; Silvestro et al., 2015). In spite of these many examples, a general framework for how to systematically constrain environmental simulations and parameter inference is still needed.

Building on the uncertainty literature across distributed model applications, we recommend constraining environmental simulations following a hierarchy of metrics/signatures and corresponding constraints (McGlynn et al., 2013). This framework builds from sources of information that should be widely available and have low cost to obtain, to information that may not be available everywhere and that may be more expensive to obtain. We present this framework in Figure 2, outlining information sources that range from regional signatures, to error metrics, to spatial patterns informed by





observations and expert judgment.  In this study, we use this framework as a path to evaluate distributed model equifinality during model calibration.

These signatures/error metrics are summarized as follows:

- **Regional Signatures:** Regional signatures, increasingly used in hydrological model applications (e.g. Yadav et al., 2007; Bloeschl et al., 2013; Hrachowitz et al., 2014), constrain key annual dynamics. Ranges for a region may be informed by global or regional data sources and publications (e.g., Krug et al., 1990; Milly et al., 1994; Church et al., 1995; Zomer et al., 2007, 2008).

- **Local signatures:** In the absence of measurements, field researchers often have a broad sense of feasible and infeasible ranges for different types (e.g., evapotranspiration, water table height/soil saturation, snow water equivalent) of annual or seasonal hydrologic behaviour. These constraints are best applied to average catchment conditions, with a goal to remove simulations that are too wet or too dry at an annual or seasonal timescale.  This information could be considered "soft information" as described by Seibert and McDonnell (2002), because it may not directly based on measurements within the catchment.

- **Error Metrics:** In general, error metrics are calculated by comparing time series of observational records (most commonly, streamflow, but can include snow water equivalent or snow depth (e.g., Whitaker et al., 2003), road ditch flow (e.g., Surfleet et al., 2010), soil moisture (e.g., Cuo et al., 2006) to model simulations. We recommend comparing observed and simulated time series data in two ways (e.g., van Werkhoven et al., 2009; Hrachowitz et al., 2014), as statistical metrics, which measure model performance with respect to the entire time series, e.g. Root Mean Squared Error (RMSE), Nash Sutcliffe Efficiency Coefficient (NSE; Nash and Sutcliffe (1970)), and dynamic metrics, which measure model performance with respect to different periods or types of hydrologic behaviour, e.g. the baseflow index, the slope of the flow duration curve (Wagener et al., 2001; Gupta et al., 2008; Pfannerstill et al., 2014; Shafii and Tolson, 2015).  Performance across statistical metrics is typically judged with respect to a threshold value, e.g., NSE greater than 0.8, or some threshold percentage, e.g., top 10% of RMSE values (e.g. Moriasi et al., 2007; Harmel et al., 2014).  Dynamic metrics may expand assessment of hydrologic behaviour, as existing work has shown that there is information contained not only in different types of data but also in different periods for an observational time series (Wagener et al., 2001; Gupta et al., 2008; Pfannerstill et al., 2014; Shafii and Tolson, 2015).

- **Internal catchment behaviour:** Aggregated spatial predictions (if values are known) and simulations (if values are unknown) may be used to assess the realism of predicted internal catchment behaviour (Grayson et al., 2002; Wealands et al., 2005; Kuraś et al., 2011; Koch et al., 2016; Fang et al., 2016). There are many examples of distributed models that have evaluated internal catchment behaviour using what Grayson et al. (2002) refer to as the 'many points' approach – comparing model simulations to (often, time series) observations at several locations throughout the catchment for a given process of interest (e.g., Thyer et al., 2004; Kuraś et al., 2011). However, in the absence of internal observations, we suggest that internal simulations may still be incorporated into evaluations





of distributed model behaviour and parameter estimation. First, we suggest researchers apply a simple 'reality check' – are the predictions possible? This step may help to identify runs that predict behaviour outside of the realm of possibility. Secondarily, we suggest that researchers should consider the parameter influences that generate differences in the spatial representations, to evaluate and either accept or reject parameter sets as more realistic or

less realistic representations of the system. This type of evaluation will benefit from involving experimentalists that have experience in a particular catchment. While this evaluation is often qualitative, it will rely on expert knowledge, previous experimental catchment studies, as well as a perceptual catchment model

### 3.2 Approach

In this study, we apply the framework demonstrated in Figure 2 within a Monte-Carlo based uncertainty analysis,

interrogating relationships between model parameter values, model simulations and predictions, and model metrics that summarize model performance and behaviour as catchment-wide signatures and error. We applied Monte-Carlo sampling to a priori constrained parameter ranges for 53 independent parameters associated with basin-wide properties, soil properties, and vegetation properties to produce 10,000 parameter samples (Table 1; Appendix A). The model was executed for each parameter sample, and model simulations were used to calculate eleven different metrics for each model run. We

acknowledge that 10,000 parameter sets does not fully explore the entire parameter space for a 53-parameter model. However, our approach here serves as a demonstrative exercise, to provide an example of how a modeller could approach this type of reduction via multiple constraints. As such, having a greater number of parameter sets should not influence our general conclusions.

Following the framework in Figure 2, Table 2 lists all metrics used to assess catchment behaviour, the equations

used to calculate each metric, the associated constraints we applied in this study, and the source of information for defining each constraint. Due to the relatively short time period for which all meteorological measurements were present within the catchment and at both SNOTEL sites, we assessed model calibration across a single year of data corresponding to the 2008 water year (WY) for one discharge location (Stringer Creek, LSC). To further demonstrate the value of this approach, we demonstrate performance for two other discharge locations, a smaller catchment (Middle Stringer Creek, MSC) and a larger

catchment (Tenderfoot Creek, LTC) for both the 2007 and 2008 water years. We focus specifically on a calibration period of the 2008 water year though results are demonstrated for two years of data.

The framework begins with an evaluation of whether model simulations match signatures of regional behaviour, evaluated via the runoff ratio and aridity index. While the latter is calculated from potential evapotranspiration (PET), which may be supplied as input for some models, PET varies with parameters in DHSVM. Signatures were constrained using

existing datasets and regional reports/publications (e.g., Sankarasubranmanian and Vogel, 2003), including the CGIAR-CSI Global Aridity and Global-PET database (Zomer et al., 2007; 2008), which was used to limit AI to values obtained within a 50 km radius of Tenderfoot Creek.





We additionally incorporated local signatures of cumulative annual evapotranspiration and average annual water table depth, applied to catchment-wide averages. Experimentalists can often recommend basic ranges for these values from fieldwork or from numerous visits or observations of other hydrologic processes across the catchment. ET was constrained based on calculations from Mitchell et al. (2015), from observations at a meteorological tower located within the catchment.

We limited annual water table depths based on previous findings that the water table is only active for a small portion of the year during snowmelt, with most of the catchment experiencing very little response (Jensco et al., 2009; Jensco and McGlynn, 2011).

Error metrics were calculated for observed streamflow measurements as well as snow water equivalent (SWE) measurements at two SNOTEL stations within the larger Tenderfoot Creek catchment. Continuous observations of snow

water equivalent are available at both SNOTEL sites. Statistical fits for simulated and observed streamflow and SWE were assessed with the NSE. We additionally selected five metrics to describe dynamic streamflow and SWE behaviour:

- Runoff ratio error ($RRE_Q$), to capture the annual water balance,
- Error in the slope of the flow duration curve ($SFDCE_Q$), assessed between the 10th ($p_l$) and 30th ($p_h$) percentile exceedance flows, to capture the variability in transition flows,

- Error in the magnitude of the streamflow peak ($P_Q$),
- Timing of the peak magnitude ($T_Q$), both selected to capture the interaction between snowmelt and streamflow, and
- Total storage error ($VOL_S$), similar to the $RR_E$ but representative of total annual SWE storage behaviour, to match overall SWE volume.

As individual daily flow events are the most difficult to match and tend to be influenced by a range of different model

parameters, we left constraints on these behaviours to be widest (Table 2). Recognizing that many of these metrics may be conflicting, our goal is to end with parameter sets that perform reasonably well for many different types of streamflow and SWE behaviour, as opposed to performing well for only one or two metrics, yielding an unrealistic hydrograph or SWE time series.

Error metrics for the SNOTEL sites were computed at each site, but presented in the manuscript as a single measure

across both sites using an 80% (Onion Park) – 20% (Stringer Creek) weighting. We chose this weighting because Onion Park elevation is more representative of Stringer Creek catchment topography, with more than 90% of the catchment at an elevation closer to the Onion Park (2259m) than to Stringer Creek (1983m) site elevations. When weighting both sites into a single metric, the absolute values of the storage volumes were averaged, as the model tended to underpredict at Stringer Creek and overpredict at Onion Park. Last, we extracted spatial predictions of water table depth for Stringer Creek at noon

each day from Oct 1, 2007 through Oct 1, 2008 across the entire catchment. These maps were averaged across different periods to create maps of average annual predictions, average predictions during snowmelt (May 1 – July 30), and average predictions during the dry period during the late summer (August 1 through September 30).





## 4 Results

### 4.1 How Do Single Constraints Influence Model Performance?

As can be seen in Figure 3, constraining with a single metric yields good performance for matching observations of SWE (with respect to NSE and error in SWE volume). However, the ranges and first and third quantiles for all other metrics

are well outside of constraints. For example, error in peak timing may exceed upwards of a month (30 days) and error in peak magnitude plus or minus 100% when the model is constrained with just one metric.

### 4.2 How do constraints on model simulations influence the number of acceptable parameter sets?

The number of parameter sets that met constraints applied individually, hierarchically, and as a group are shown in Figure 4. When applied hierarchically, a total of 9 parameter sets met all metric constraints (Figure 4a). Individually, a large

number of parameter sets met behavioural constraints on SWE NSE, removing only 489 of the 10,000 sets. The opposite was true for behavioural sets constrained based on peak streamflow, which identified only 626 behavioural parameter sets.

Figure 4(b) summarizes the number of acceptable parameter sets that meet groups of constraints, as introduced in Figure 2. Across the different types of signatures and metrics, dynamic constraints overwhelmingly had the single greatest impact on reducing acceptable parameter sets, with just 26 sets meeting all criteria for dynamic metrics. Interestingly,

statistical and regional constraints identified a similar number of behavioural sets. While combining regional and local metrics reduced the number of behavioural sets, combining statistical and dynamic constraints did not reduce the number of behavioural sets.

### 4.3 Are All Metrics Needed?

To evaluate whether all eleven metrics and corresponding constraints are needed to ensure behavioural consistency

within this framework, we examined model behaviour and performance across subsets of metrics (n = 2 to 10) as well as the level of redundancy across metrics (Appendix B). Employing only two or three metrics removes 69.7% and 74% respectively of the sets removed when all 11 metrics are used, suggesting that 25 to 30% of parameter sets inconsistent with model behaviour across our framework would remain. By comparison, including eight metrics generally removes an average of 92.6% of all nonbehavioural parameter sets. Of the 165 combinations of eight different metrics evaluated in this analysis,

83% included at least one metric in each of the different framework groups (regional, local, statistical, dynamic). The redundancy of each metric was also evaluated by comparing the nonbehavioural sets identified via constraining one metric that would not be found by another (Appendix B). While we found redundancy did occur in this type of framework, more than half of the simulations found to be nonbehavioural by constraining one metric were considered behavioural by another metric across a majority of metric combinations.




**4.4 How Do Single and Multiple Constraints Impact the Distributions of Metrics of Catchment Behaviour?**

Initial distributions (I) of metric values across all 10,000 parameter sets include a wide range of behaviour (Figure 5). While these distributions can be narrowed by the addition of one constraint, many distributions of behavioural performance were also narrowed by the addition of all constraints. The addition of all constraints favoured higher values (~0.5 to 0.7) for the runoff ratio and lower values (~0.35 to 0.5) for the aridity index and average water table height (~-0.75 to -0.95). Performance improved slightly for both statistical metrics ($NSE_Q$ and $NSE_S$), and narrowed ranges for both error in the runoff ratio ($RRE_Q$) and slope of the flow duration curve ($SFDCE_Q$). While behavioural sets both over- and underestimated $RRE_Q$ and $SFDCE_Q$, the final behavioural sets only included simulations that underestimated peak streamflow alongside earlier than observed peak timing, while overpredicting SWE.

**4.5 How Do Constraints Impact Our Certainty in Predictions and Simulations of Catchment Behaviour?**

Figure 6 displays the impact of different constraints on the range of model simulations for streamflow (LSC), snow water equivalent (Stringer Creek SNOTEL), and average water table depth for the 2007 and 2008 water years. Streamflow simulations for the calibration period (WY 2008) were best narrowed by the addition of statistical and dynamic constraints and poorly narrowed by the addition of regional and local constraints. Constraining behaviour via regional and local metrics resulted in behavioural simulations that underestimated peaks and poorly matched timing of streamflow initiation and maximum response. While statistical and dynamic metrics both predict an earlier streamflow response on the rising limb of the hydrograph, simulations well-matched behaviour on the falling limb of the hydrograph. Dynamic metrics also better constrained streamflow simulations during baseflow, which tended to be overestimated by behavioural sets selected by statistical metrics.

Predictive ranges for SWE were similar across different constraints. All metrics well constrained SWE behaviour (shown for the Stringer Creek SNOTEL station), with consistent differences between simulations and observations regardless of metrics used to constrain behaviour. SWE magnitude was overestimated for all behavioural sets, though the initiation of melt timing was similar, producing simulations that overestimated the time that the last of the snowpack melts during late May and early June.

In contrast to simulations of SWE, behavioural simulations of average water table depth varied widely depending on which metrics were used to constrain behaviour. Behavioural sets identified via statistical metrics included a wide range of average water table behaviour, suggesting that statistical metrics may poorly inform water table dynamics. Behaviour constrained by local and regional metrics produced a pattern of average water table depth consistent with general understanding of water table behaviour (Jensco et al., 2009; Jencso and McGlynn, 2011). As a metric for average water table depth was directly incorporated into the assessment of regional and local behaviour, this result is expected. Dynamic metrics had the largest impact on narrowing WTD simulations, identifying behavioural simulations that were most consistent with our understanding of water table response across Tenderfoot Creek.





**4.6 How Well Do Model Simulations Match Observations?**

Calibration to the 2008 water year utilizing all metrics and corresponding constraints identified behavioural sets corresponding to hydrographs and SWE time series that reasonably matched observations (Figure 8). Simulations of streamflow at the calibration site (LSC) for the 2008 water year simulated the double streamflow peaks as well as the relative

timescales of wetting up and drying down that occurred in the observational record, with some differences between simulations and observations. However, the timing of streamflow on the rising limb of the hydrograph and the first streamflow peak were poorly matched. SWE at both SNOTEL sites was overestimated in terms of magnitude and timing, though simulations generally matched observations. Melt timing generally approximated observed behaviour at Onion Park but was slightly early at Stringer Creek, though again dynamics in both cases are largely matched, especially the timing of

accumulation and melt from April through June.

To demonstrate the value of this approach with respect to other streamflow observation locations and for years beyond the calibration period, we also display fits to the 2008 water year for Middle Stringer Creek and Lower Tenderfoot Creek. Streamflow NSE at these two locations varied between 0.63 and 0.83 for LTC and between 0.47 and 0.74 for MSC. Like fits to Stringer Creek, timing and peak magnitude for streamflow at MSC and LTC were underpredicted with respect to

observations, though simulations approximate behaviour of observations on the rising and falling limbs and accurately predict the presence of observed double streamflow peaks for the 2008 water year. Performance for other metrics for these two discharge locations is reported in Appendix D.

Outside of the calibration period, fits for the 2007 water year for all locations were high (Figure 8). NSE for the calibration site (LSC) varied between 0.65 and 0.82, while NSE for the other two streamflow locations varied between 0.64

and 0.91 at LTC and between 0.51 and 0.78 at MSC. While the water balance ($RRE_Q$) was well approximated for LSC, estimates were higher than observations for both MSC and LTC, though timing and magnitude of peak streamflow were well approximated. The values for all metrics are reported in Appendix B.

**4.7 How Do Constraints Impact Parameter Certainty?**

To test whether multiple model constraints on hydrologic behaviour were able to reduce parameter equifinality, we

investigated the extent to which constraining model-predicted and simulated behaviour narrowed ranges for parameter values, with results displayed in Figure 8. While ranges remained wide for many parameters, ranges were greatly narrowed for several of the 53 model parameters. Narrowed ranges were observed especially for lateral conductivity (7) and its exponential decrease with depth (8), maximum (26) and minimum (27) understory stomatal resistance, and a scalar on overstory LAI (50). We applied a two-sample Komolgov-Smirnov test to assess whether there was a statistically significant

difference ($p < 0.1$) between the original distribution of parameter values and the distribution of parameter values for sets that met framework criteria. We found that all but five parameters exhibited statistically significant differences between their original and final value distributions. However, ranges were still wide for many parameter values.





**4.8 Do Assessments of Catchment Averages and Observations Produce Consistent Internal Predictions of Catchment Behaviour?**

In this study, we do not directly compare simulations of water table depth to well observations. Instead, we sought to assess the variability across these simulations for three different time periods, to determine whether behavioural parameter sets yielded similar simulations, and to ask whether spatial diagnostics beyond time series observations should be incorporated into assessments of distributed model behaviour. All predictions of water table depth are included in Appendix B, with three representative simulations of water table depth corresponding to the driest, intermediate, and wettest simulations shown for annual, snowmelt (May 1 – July 31), and late summer dry-down (Aug 1 – Sept 30) periods included in Figure 9. Seasonally, water tables were simulated closer to the surface during snowmelt and closer to bedrock during late summer, with average behaviour somewhere between these two extremes.

**4.8.1 Are Simulations of Water Table Depth Consistent in Space and Time?**

Across simulated water table depths for nine behavioural parameter sets, differences in simulations were large in both space and time (Figure 9; Appendix B). At annual timescales, many parts of the catchment were predicted to have similar annual average behaviour, with differences below 0.1m for 38% of all cells. Simulations across 13.3% of the catchment differed by more than 0.2m, 20% of the modelled soil depth. These numbers were comparable for simulations during late summer, when the majority of the catchment was simulated to have similar behaviour. Differences were greatest during the snowmelt period, exceeding simulated ranges of 0.2m over more than 64% of the catchment. The locations of the largest differences also varied across seasonal and annual periods. Our results show that equifinality can produce vastly different simulations of internal catchment behaviour.

**4.8.2 Can Simulations Be Used to Further Reduce Equifinality?**

Evaluating whether simulations of internal behaviour of water table depth match perceptions and observations of catchment functioning may be done using simple metrics of spatial behaviour. One such example is the presence of the water table at the land surface. Tenderfoot Creek field researchers suggest that there are few locations and few times where the water table should be at the land surface across the catchment (Jensco et al., 2009; Jensco and McGlynn, 2011). Based on these recommendations, we expect high water tables to be present across minimal areas (<5%) of the catchment. To test the ability of a spatially distributed high water table metric to discern differences between behavioural sets, we calculated the percentage of cells across Stringer Creek where the water table was simulated to be within 0.05m of the surface for the entire snowmelt period (May – July). Four different behavioural sets simulated high water tables over more than 5% of catchment; at the extreme end, one set simulated high water tables over 24% of the catchment. Five sets simulated high water tables over less than 5% of the catchment, with simulations below this threshold across the entire catchment for four of these sets. We would conclude from this evaluation that these four sets produce spatial behaviour that is consistent with experimental insight across Stringer Creek.




## 5 Discussion

Within this study, we develop a conceptual uncertainty analysis and framework for parameter uncertainty reduction with application to distributed modelling of headwater systems. We demonstrate how this framework can be applied with respect to a case study in Stringer Creek, a headwater catchment located in Montana. While there are a few examples of
frameworks for model calibration (e.g. Refsgaard, 1997) as well as several studies that have evaluated the tradeoffs between model complexity and predictive uncertainty, there remain few guidelines for specific use in distributed hydrologic models, which have their own challenges. In this study, we sought to create and test a framework that considered many of the above discussed limitations and capitalized on the strengths of such complex models.

### 5.1 The Value of Suites of Metrics in Distributed Model Applications

Many distributed model applications use only one constraint to select behavioural parameter sets or to justify model performance. As we show in Figure 3, model simulations that meet just one set of criteria may poorly match other catchment-wide basin behaviour. Of particular interest is the impact of other constraints on catchment-wide signatures, including the runoff ratio, aridity index, annual evapotranspiration, and annual average water table depth. For our application, applying a single statistical or dynamic metric did not narrow any signature to ranges deemed to be
representative of regional or catchment behaviour.

Interestingly, despite the snow-driven nature of Tenderfoot Creek, many parameter sets were equifinal with respect to the NSE for SWE (Figure 5). This suggests, for this particular application, that a priori parameter ranges did not generate wide variability in SWE time series at the two observation sites, leading us to believe that within DHSVM the meteorological forcing data is a greater driver of modelled SWE. Since this forcing data is also uncertain, there is potential
for this uncertainty to propagate into model simulations, and may be driving the overestimation of SWE observed at both SNOTEL sites. Previous model analysis in this catchment found that many snow accumulation and melt parameters were insensitive and highly interactive (Kelleher et al., 2015). Together, these results broadly suggest that certain types of internal behaviour, especially point observations, may not always reduce equifinality or improve predictive certainty. Other studies have questioned the empirical equations used to calculate SWE accumulation and melt, and have found improvements in the
prediction of SWE accumulation and melt by altering hard-coded parameters within the model framework (Thyer et al., 2004; Jost et al., 2009).

The largest number of nonbehavioural sets was identified by error in peak streamflow magnitude. While there were several sets that had very small errors in peak streamflow magnitude (Figure 5), many overestimated the annual water balance, and were therefore identified as nonbehavioural. While peak timing was underestimated by the addition of all
eleven criteria, error in the runoff ratio and the slope of the flow duration curve were narrowed to ranges of lower error and average water table depth was constrained to more representative ranges when all eleven criteria were added (Figure 6).





In the absence of time series observations, our results show that regional and local metrics could be used to narrow predictive certainty, but may poorly match hydrologic behaviour for some years (Figure 6). While statistical or dynamic metrics may narrow predictive certainty for streamflow in similar ways, we found that they informed simulations of other key catchment behaviour differently. Statistical constraints poorly constrained simulations of water table depth (Figure 6).

In contrast, dynamic constraints, applied only to errors in streamflow and SWE time series, yielded behavioural simulations that closely matched expectations of average water table behaviour across the catchment (Jencso et al., 2009; Jensco and McGlynn, 2011). Our results demonstrate that suites of metrics related to hydrologic behaviour may inform simulations of other hydrologic processes (i.e., average water table depth). In contrast, we found statistical metrics to carry little information regarding other types of simulated or predicted behaviour (Figure 3, Figure 6). To our knowledge, most

uncertainty analysis studies with semi-distributed or distributed models typically favor statistical constraints over dynamic constraints (e.g., Shields and Tague, 2012; Safeeq and Fares, 2012). It is unclear whether this result is generalizable to other catchment applications, though we expect the impact of dynamic metrics with regards to discerning catchment behaviour to only increase for a greater number of hydrologic events.

**5.2 Benchmarking Simulations Against Other Model Applications to Stringer Creek**

Altogether, application of a framework for distributed modelling yields model simulations that match streamflow at the calibration site and that generally match patterns of SWE accumulation and melt (Figure 7). In general, while results for the 2008 water year bracket observations, results for the 2008 water year are underpredicted with expect to peak behaviour and timing. Given that snowmelt drives both rising limb and peak hydrograph response, uncertainty related to solar radiation or air temperature forcings may be driving the difference in rising limb behaviour we found in our simulations.

Simulations compare favorably to other model applications to Stringer Creek. Nippgen et al. (2015) developed a parsimonious distributed model, with application to Stringer Creek yielding NSE values of 0.8 for WY 2007 and 0.94 for WY 2008. Smith et al. (2013) developed the catchment connectivity model (CCM), a conceptual hydrologic model that predicts streamflow based on relationships between terrain, connectivity between the hillslope and stream found in Jensco et al. (2009), and the duration of flow. Simulation of Stringer Creek with CCM achieved similar levels of fit to our study (NSE

of 0.81 for box-cox transformed streamflow; Smith et al., 2013), as did model simulations of LTC (NSE of 0.903) and MSC (NSE of 0.856; Smith et al., 2016). Simulations of streamflow by Nippgen et al. (2015) and Smith et al. (2016) also underestimated peak streamflow for the 2008 water year, suggesting that uncertainty in the meteorological forcing data may be responsible for poor performance during calibration period snowmelt. Ahl et al. (2008) modelled Tenderfoot Creek streamflow using the soil and water assessment tool (SWAT) for the period 1995 – 2002, and achieved an average NSE of

0.86 during calibration and 0.76 during validation. Our best fit by NSE to Stringer Creek was 0.79 for the 2008 water year and 0.82 for the 2007 water year, while best fits for Tenderfoot Creek were 0.83 for the 2008 water year and 0.91 for the 2007 water year. Thus, the level of fit we achieved through this investigation was similar to other modelling of this area. Moreover, we ensure that the selected model runs also match key catchment-wide behaviour as well as dynamical





streamflow behaviour through the use of additional metrics. We acknowledge, however, that sources of uncertainty beyond those considered by our study may still be driving differences between simulations and observations. Possible sources of error in simulations of both SWE and peak magnitude and timing errors may be related to uncertainty in the model framework, model forcing data, and observations used to judge performance. Future work modelling these catchments will

seek to address these other sources of uncertainty alongside uncertainty in parameter values.

### 5.3 Parameter Uniqueness and Equifinality

We are often interested in how constraints influence predictive uncertainty of hydrologic behaviour. Thus, the logical follow-up question is whether these constraints help to narrow values for model parameters. Across the 53 model parameters included in our analysis, constraints had a minimal impact on narrowing parameter ranges for most parameters.

However, we did detect a subset of soil and vegetation parameters that were reasonably narrowed from their original ranges. This suggests that equifinality is still present, but that uncertainty analysis may also improve parameter certainty. In a similar application of DHSVM to the Oak Creek catchment near Corvallis, OR, Surfleet et al. (2010) also found significant equifinality when using a similar approach to characterize uncertainty with respect to streamflow and road-ditch flow prediction. While we cannot formulate any strong conclusions regarding predictive uncertainty in parameter values given our

limited sampling of the parameter space, we assert that equifinality may overwhelm the ability to extract much information regarding parameter values in complex distributed model applications. However, equifinality with respect to catchment average conditions may manifest in variable predictions of internal catchment behaviour (e.g., Figure 9). Thus, evaluating spatial predictions may be one of the few practical approaches to reducing this equifinality.

### 5.4 Interpretation of Internal Behaviour

Both field investigations and modelling at Tenderfoot Creek have focused on observation and prediction of water table response with the goal of improving our interpretation of streamflow generation and the connection between rainfall and runoff (Jencso et al., 2009; Jensco and McGlynn, 2011; Nippgen et al., 2015). Thus, we chose to evaluate simulations of water table depth for our case study. We found that parameter sets that were equifinal with respect to streamflow and SWE (Figure 7, 8) produced vastly different annual and seasonal simulations of water table depth (Figure 9). Moreover, the

locations where simulated differences were largest and smallest across the catchment also varied with time. By assessing the fraction of the catchment simulated to be at or near the surface across the snowmelt period, we found that four of the nine sets produced near-surface water tables over a larger area than previous work suggests is likely. In this sense, internal behaviour can be used to identify simulations that do not match perceptions or direct observations of catchment behaviour beyond comparison with time series. This approach enables one to move beyond just matching to a few points with

observations, encouraging modellers to holistically evaluate performance and simulation of multiple hydrologic processes.





### 5.5 The Case for Adding Spatial Predictions

Although constraints on the hydrograph and other hydrologic behaviour may ultimately match observations, we are still faced with the likelihood that parameter equifinality may not be eliminated by matching predictions to a few time series observations. Aggregating model predictions of spatial patterns is time intensive but can be highly informative. Therefore,

we recommend mapping these patterns once regional values and/or observations have already been used to narrow the parameter space.

Distributed model predictions of internal behaviour are often performed by matching model predictions of different catchment variables (e.g., snowmelt, Thyer et al., 2004; groundwater table dynamics, Kuras et al., 2011) to multiple observations at discrete points across a given catchment, though several studies have also shown how patterns of hydrologic

processes may be directly incorporated into evaluating and applying complex environmental models (Grayson et al., 2002; Wealands et al., 2005; Fang et al., 2016; Koch et al., 2016). In the absence of time series observations of other hydrologic processes, simulated patterns may be evaluated using expert knowledge, understanding of process controls across landscapes, and other information knowledge and experience about model output realism (e.g., Franks et al., 1998). In our example, nine parameter sets that matched the hydrograph and associated SNOTEL station SWE observations equally well

(Figures 5, 6, and 7) predicted very different patterns of internal catchment behaviour (Figure 9). However, these same sets also exhibited divergent parameter values for several different soil and vegetation properties (Figure 8). This result exemplifies the additional information available to constrain behavioural sets as well as the potential for catchment-scale predictive uncertainty driven by parameter equifinality. Sets that met framework criteria simulated very different patterns of water table depth across both annual and seasonal periods (Figure 9). Differences typically were between 0 and 0.2 across

all simulations and all catchment cells, but were especially large (> 0.5m) over small fractions of the catchment. If spatial predictions were not used to limit the parameter sets, any one of the nine sets has potential to propagate predictions for future land use or climate change scenarios that would lead to vastly different expectations of water table presence across the landscape. We conclude that equifinal sets may generate very different simulations of internal catchment behaviour, and therefore recommend that spatial simulations should be incorporated into assessments of distributed catchment behaviour

(Figure 2). While we chose to highlight only one spatial diagnostic in this paper, we advocate the inclusion of multiple diagnostics related to key catchment storages (e.g., water table depth) and fluxes (e.g., evapotranspiration).

### 5.6 Future Uncertainty Analysis of Distributed Catchment Models

Applying this framework to other catchments will require researchers to select a set of metrics for assessing model performance, emphasizing both temporal and spatial metrics that may help to ensure appropriate representation of key

catchment processes (e.g., Yilmaz et al., 2008). Evaluating any of these aggregated signatures, metrics, and spatial diagnostics should be based on a strong conceptual understanding of the catchment and how processes that govern the water





balance change temporally and spatially at multiple scales. Such evaluations may especially benefit from joint evaluation by modellers and experimentalists (Seibert and McDonnell, 2002).

The approach we recommend in this manuscript builds on and compliments several recent studies that have sought to improve process consistency across models of varying complexity and within distributed hydrologic models. Many recent
studies have shown that, despite their weaknesses, distributed models typically outperform conceptual models with respect to reproducing signatures (e.g., Hrachowitz et al., 2014) and matching hydrograph dynamics (e.g., Euser et al., 2015). Thus, the new objective we face is how to improve our approaches to distributed modelling, ensuring model realism while minimizing uncertainty. Work by several researchers has evaluated methods for the spatial distribution of parameter values, to ensure process consistency across catchment scales (Euser et al., 2015; Fenicia et al., 2016; Nijzink et al. 2016). Others
have sought to incorporate expert knowledge to limit the feasible parameter space (Gharari et al., 2014; Nijzink et al. 2016). However, as shown by Surfleet et al. (2010), equifinality may still overwhelm our ability to draw meaningful conclusions from distributed data, especially at small headwater scales. Taken together, these studies suggest that equifinality is likely to still limit application of distributed models, but that carefully evaluating how this equifinality may impact uncertainty in predictions and simulations alongside parameter values may enable more careful use of distributed models. Similarly, these
studies also suggest that incorporating constraints within distributed model frameworks based on expert knowledge and alternative data sources, whether applied to model output, model setup, or model parameter values, may ensure more holistic process representation across a given catchment.

There are few studies that have sought to characterize equifinality and uncertainty for physically-based, distributed model applications, but the number of distributed model applications that either incorporate uncertainty is growing. Of those
that exist, most have focused on characterizing predictive uncertainty in terms of uncertainty in parameter values (e.g., Cuo et al., 2011; Shields and Tague, 2012; Tague et al., 2013), or in terms of model framework uncertainty, utilizing fixed parameter values and modifying the model formulation to match multiple experimental observations throughout the critical zone (Thyer et al., 2004). Alternatively, there is also a body of work that treats the model framework itself as a form of uncertainty, testing different model structures as hypotheses for how a catchment may function (Clark et al., 2011; Fenicia et
al., 2011; Hrachowitz et al., 2014). This approach may also provide an alternative to predicting hydrology via a model with fewer parameters than the distributed application shown here, with a model structure that incorporates the level of detail mandated by the complexities of the catchment. These studies exemplify the common need to consider uncertainty when predicting environmental behaviour with complex, many parameter models. Our work suggests this will only provide the modeller with a better understanding of the catchment but also of the model in question. Altogether, research with
distributed models and our own analysis do critique some of the challenges associated with distribute model application, but also highlight the value of distributed models for hydrological predictions (Hrachowitz et al., 2014; Fitachi et al., 2016).





**6 Conclusions**

Distributed, complex models are powerful tools that enable exploration of spatial and temporal simulations and future scenarios of alteration. While beneficial, their complex nature and subsequent potential for equifinality calls into question the typical process researchers use to achieve a representative set of parameters for a given application. We
performed a modelling study for a headwater catchment, Stringer Creek, located in Tenderfoot Creek Experimental Forest in central Montana, and evaluated simulations using observational records, expert insight from Tenderfoot Creek researchers and existing publications, and regional datasets. In this application, we demonstrate a method to evaluate how constraining model predictions via hydrologic signatures, model error, and process insight impacts predictive and parameter certainty, the size of the parameter set space, and potential for equifinality. Constraints include those that have potential to be available
everywhere, based on both regional datasets and local knowledge, and those that are based on observational records of hydrologic behaviour. We also include evaluation of spatial patterns of model predictions, to evaluate how parameter sets that match point observations predict storages and fluxes across the landscape.

Across all types of metrics and constraints, applied either hierarchically or in smaller subsets, we found dynamic constraints on annual, seasonal, and event behaviour to be most important for reducing predictive uncertainty and selecting
behavioural parameter sets. This suggests that researchers should use care when utilizing only statistical metrics to judge model performance or to select behavioural parameter sets, as for our application we found many model runs that had high statistical metric performance poorly matched dynamical hydrologic behaviour. Despite the large reduction resulting from applying all constraints hierarchically, nine parameter sets met all criteria. Thus, we expect that there is likely a point at which observational records and regional datasets may no longer be able to reduce parameter sets and subsequent
equifinality.

It is worth noting that parameter set selection for distributed catchment models is often done without considering whether the predicted internal behaviour of the catchment is even within the realm of reality for a site. Here, we recommend the evaluation of internal catchment behaviour as a final diagnostic to arrive at a subset of parameter sets that represent time series observations at a few locations as well as internal catchment behaviour. While evaluating average catchment
behaviour and time series observations can be helpful, these types of behaviour can often mask spatial variability of simulations across the year. Our evaluation of spatially-distributed annual and seasonal water table depth revealed somewhat consistent average behaviour but considerably variable spatial behaviour.

Overall, this approach relies on a fundamental understanding of the hydrology that governs a given area, and is only improved by adding qualitative experimental insight. Transferring our approach to other locations creates the opportunity for
a close interaction between experimentalists and modellers (e.g. Seibert and McDonnell, 2002), given that the value of a specific observations or insights to condition hydrologic models will vary widely. More than any single approach, we are advocating increased evaluation of distributed catchment models as a step towards improved representation and informed use, to ensure that spatio-temporal questions are resolved with spatio-temporally-vetted answers.



**Code Availability**

DHSVM model code is freely available at <http://www.hydro.washington.edu/Lettenmaier/Models/DHSVM/>.

**Data Availability**

Data sets were provided by the United States Forest Service and Tenderfoot Creek Experimental Forest researchers and by
NSF support to Brian McGlynn.  Stringer Creek streamflow data may be obtained from the US Forest Service at a 15-minute
resolution  from  (http://www.fs.usda.gov/rds/archive/Product/RDS-2010-0003.2/).    Aridity  data  was  obtained  from  the
CGIAR-CSI Global-aridity and Global-PET database, available at (http://www.cgiar-csi.org/). SNOTEL datasets for stations
Stringer       Creek       (Site      #       1009)      and       Onion       Park       (Site      #       1008)       are       available       at
([http://www.wcc.nrcs.usda.gov/snow/snow_map.html](http://www.wcc.nrcs.usda.gov/snow/snow_map.html)).    ALSM  data,  including  topography  and  vegetation  height,  for
Tenderfoot Creek are maintained by OpenTopography ([http://www.opentopography.org/](http://www.opentopography.org/)).

**Appendices.**

**Appendix A. Model Initialization and Parameterization**

Figure A1 and Table A1 display information used to inform the model framework and setup.  Figure A1 displays the impact
of three different hypothetical initial conditions on model predictions and simulations across nine different parameter sets.
The nine parameter sets shown in Figure A1 are the same sets that meet all criteria across the imposed framework.  This
analysis was performed to determine when the impact of initial conditions dissipated, to justify that a six month warm-up
period may be sufficiently long enough for the manuscript application.

**Appendix B. Additional Reporting of Model Results and Spatial Pattern Predictions**

To demonstrate additional analyses that were undertaken within the uncertainty analysis framework, we have included an
assessment of redundancy, additional reporting of errors for all gage locations, and predictions of water table depth for all
equifinal parameter sets.  Our analysis included eleven different metrics.  To assess whether these metrics provide unique
information to the uncertainty analysis, we tested the information contained in different combinations of metrics (Figure B1).
Figure B1 displays number of parameter sets that would be removed by all different combinations of metrics, comparing
these combinations for different subsets of metrics (2 metrics, 3 metrics, etc.).  The number of nonbehavioural model runs
identified by these different combinations are shown as distributions for each subset of metrics (2 metrics, 3 metrics, etc.),



and visualized as a percentage of nonbehavioural runs as compared to the number of nonbehavioural model runs identified by the full eleven criteria.

We also benchmarked the redundancy of metrics and their corresponding constraints within the framework and application to Stringer Creek (Figure B2). Figure B2 shows the percentage of sets removed by one metric (shown in the x-axis) that

would not be removed by another metric (shown on the y-axis). We found that one metric, error in peak flow magnitude, tended to remove nonbehavioural sets identified by other metrics. Similarly, we found that our statistical metric (NSE) may be more informative than our dynamic metric (VOL, error in SWE volume) for matching SWE time series and identifying nonbehavioural simulations. In spite of these redundancies, many metrics do provide unique information, demonstrating the value of a multi-objective approach to uncertainty reduction.

Our analysis was applied to streamflow observed at the outlet of Stringer Creek (LSC) during the 2008 water year, but metrics were extracted for two additional gage locations (Lower Tenderfoot Creek, LTC; Middle Stringer Creek; MSC) and for two years of complete simulation. Table B1 reports error metrics for the three gaging sites for both the 2007 and 2008 water years for nine behavioural parameter sets. Finally, we also include all water table depth predictions for nine equifinal parameter sets included in our analysis. Figure 9 displays a subset of predictions.

**Competing Interests.**

The authors declare they have no conflight of interest.

**Acknowledgements.**

This work was supported by NSF EAR 0943640, 0837937, 0404130, and 0337650 awards to Brian McGlynn and in part through instrumentation funded by the National Science Foundation (grants OCI-0821527 and NSF EAR 8943640, 0943640,

and 1356340). This work was additionally partially supported by the Natural Environment Research Council (Consortium on Risk in the Environment: Diagnostics, Integration, Benchmarking, Learning and Elicitation (CREDIBLE); grant number NE/J017450/1).

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





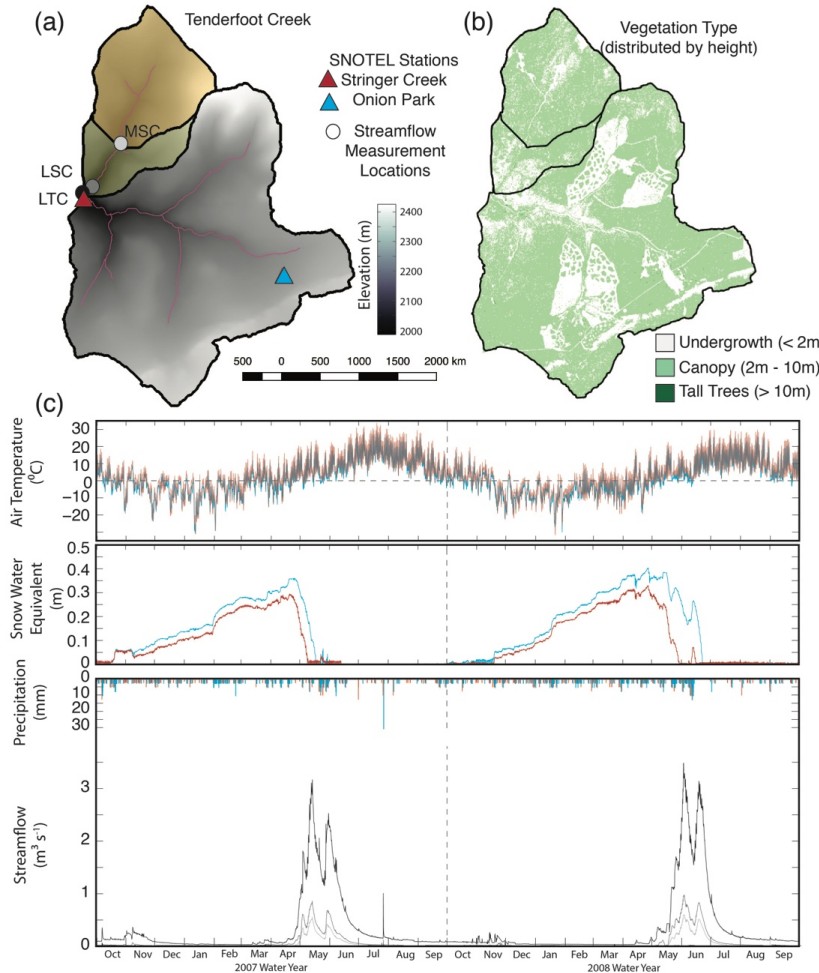

**Figure 1: Tenderfoot Creek Experimental Forest (a) DEM and instrumentation used in the study and (b) vegetation type used by DHSVM to distribute model parameters. (a) Instrumentation includes a streamflow calibration station at the outlet of Stringer Creek (LSC) as well as streamflow gaging stations at the outlet of Tenderfoot Creek (LTC) and Middle Stringer Creek (MSC). Most forcing data are measured at SNOTEL sites Stringer Creek and Onion Park. Water balance time series are extracted at both SNOTEL locations, as well as streamflow behaviour at the Stringer Creek outlet. (b) Vegetation type is distributed based on vegetation height, with cells with undergrowth vegetation shown in white, cells with canopy vegetation shown in light green, and cells with tall tree vegetation shown in dark green. (c) The model is forced with air temperature and precipitation data collected at the two SNOTEL locations.**



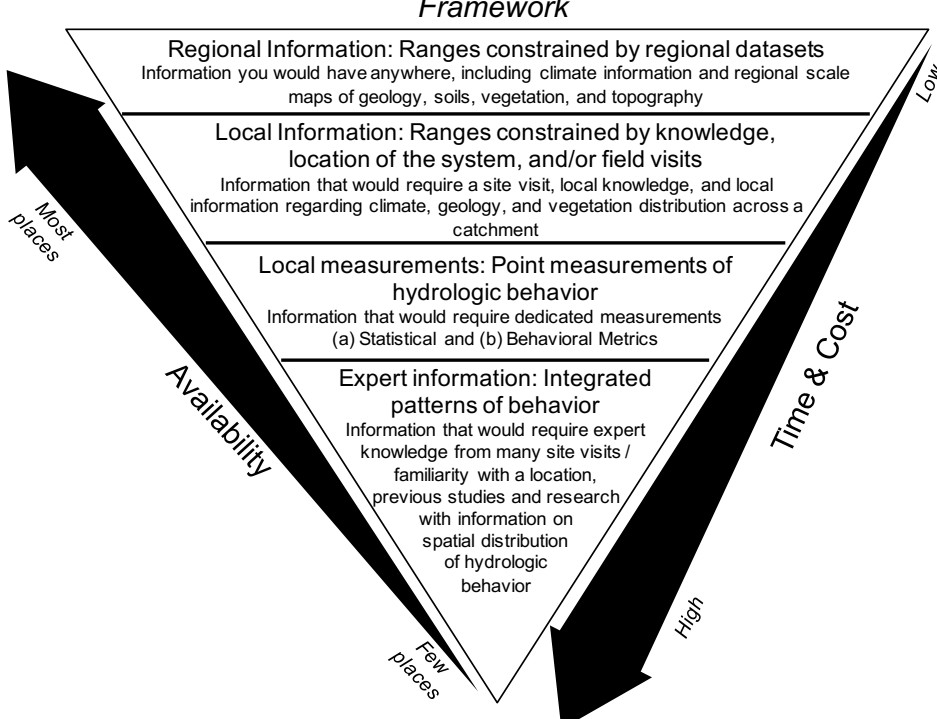

**Figure 2: Conceptual framework for constraining environmental predictions for application to a distributed hydrologic model. The framework begins with criteria based on information that has potential to be widely available and relatively inexpensive to acquire, to criteria based on observations, ending with criteria based expert knowledge that may not be available or achievable everywhere.**





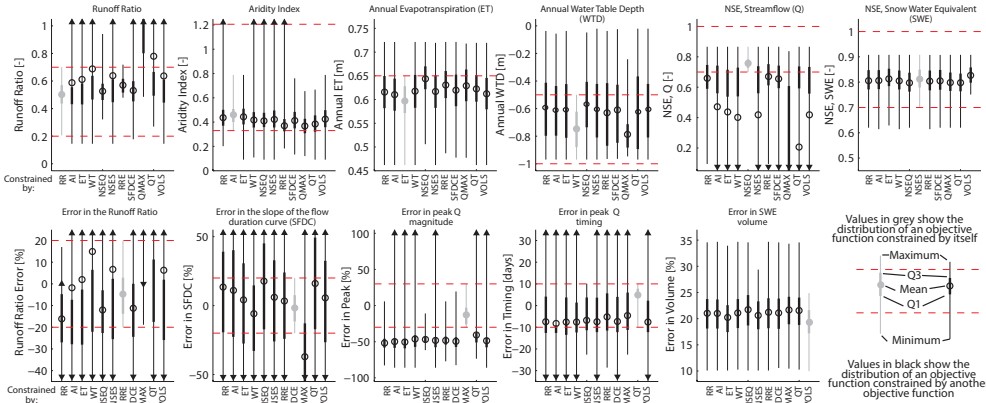

**Figure 3: Ranges, interquartile ranges, and mean values for distributions of regional and local signatures, statistical metrics, and dynamic metrics used in the proposed framework. Distributions are shown when a given metric is constrained to behavioural ranges and when a metric is constrained by other metrics, compared to behavioural ranges set in this study. Arrows indicate that**
5 **ranges exceed the visualized axis limits.**





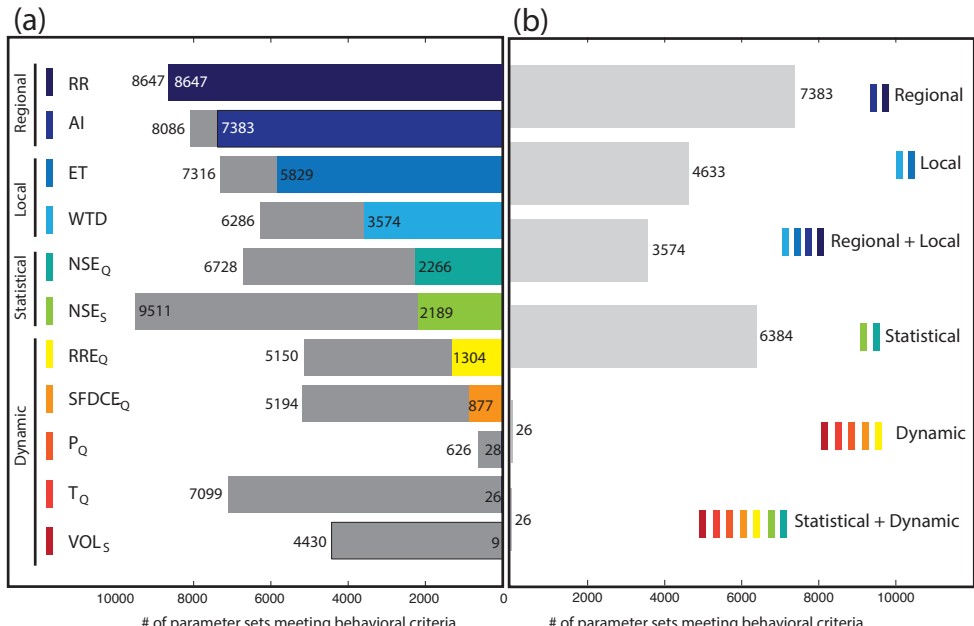

**Figure 4: Number of parameter sets removed by each stage of the conceptual framework, shown for (a) individual metrics and (b) groups of metrics specified in Figure 1. For individual metrics (a), bars in color indicate the number of parameter sets removed by sequential application of each of the steps, moving from top to bottom. Bars in grey indicate the number of parameter sets**
5 **removed if each constraint is applied independent of the others. For groups of constrains (b), bars indicate the number of parameter sets that meet all constraints in a given 'group'.**




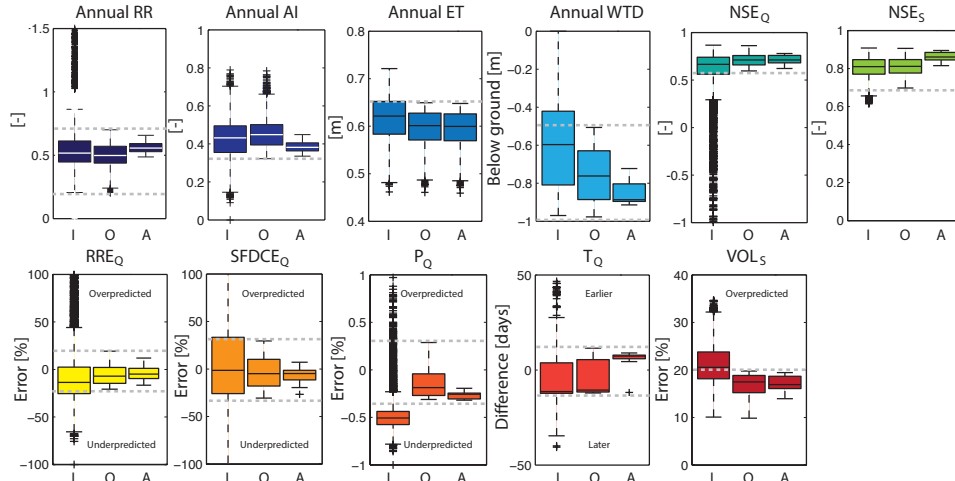

**Figure 5: The distribution of model-predicted behaviour across initial distributions of all behaviour (I), distribution of values that meet a given behavioural constraint (O), and distribution of values that meet all eleven constraints (A). Constraints are indicated by grey dashed lines. Colors correspond to different metrics outlined in Figure 3.**



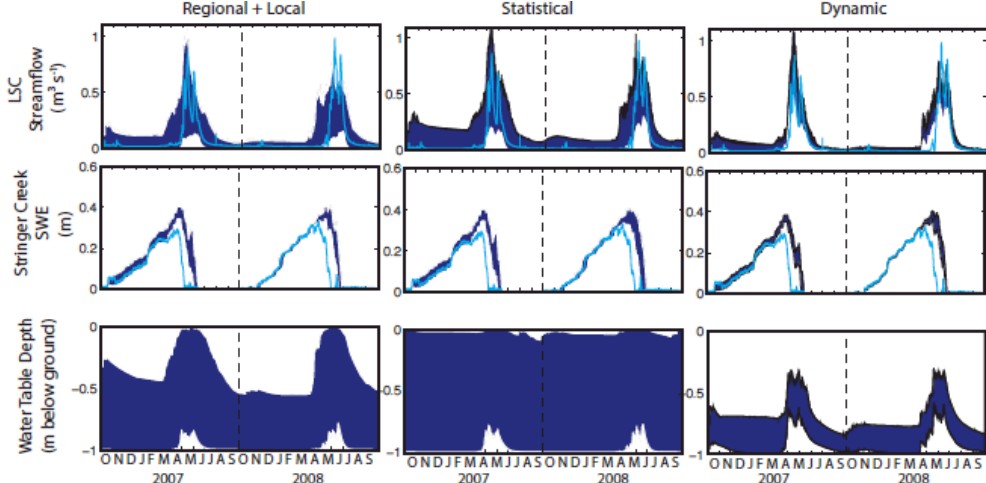

**Figure 6: Ranges and distributions for predictions of streamflow, snow water equivalent (Stringer Creek SNOTEL site), and average water table depth for grouped constraints, which are outlined in Figure 4. Predictions that meet a given set of constraints are shown as ranges in blue as a function of time. Results are shown for the calibration period (2008 WY) as well as Black dotted lines delineate the minimum and maximum predicted values across all simulations meeting corresponding constraints. Results for the full analysis are shown in (a); results generated when only sensitive parameters are sampled are shown in (b).**




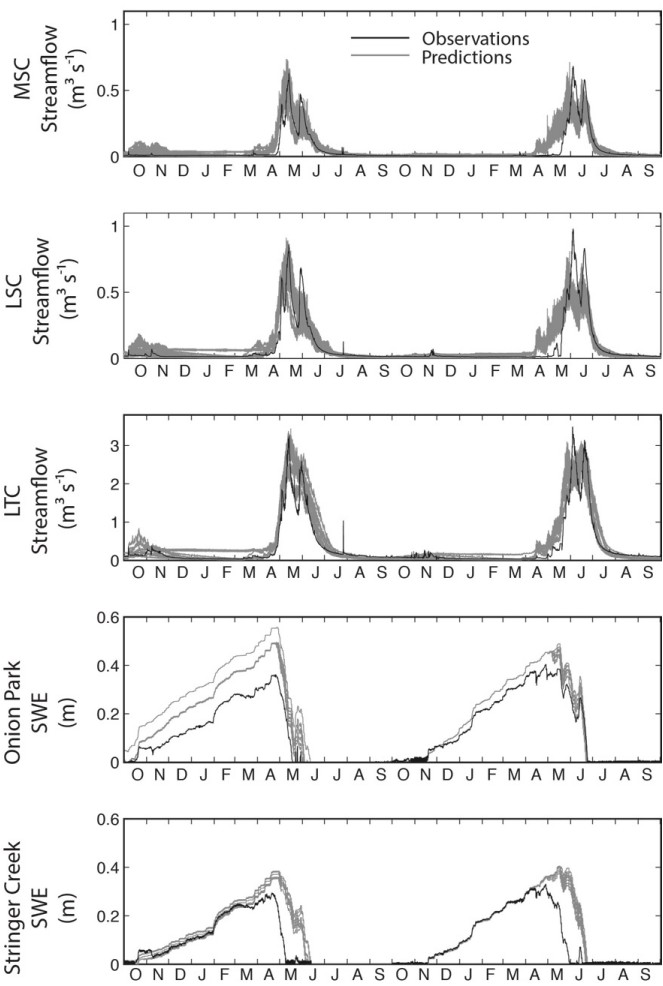

**Figure 7: Plots of model-predicted streamflow at the outlets for Middle Stringer Creek, Stringer Creek (calibration site), and Tenderfoot Creek alongside predictions and observations of snow water equivalent at the Onion Park and Stringer Creek SNOTEL that satisfy all hierarchically applied constraints.**





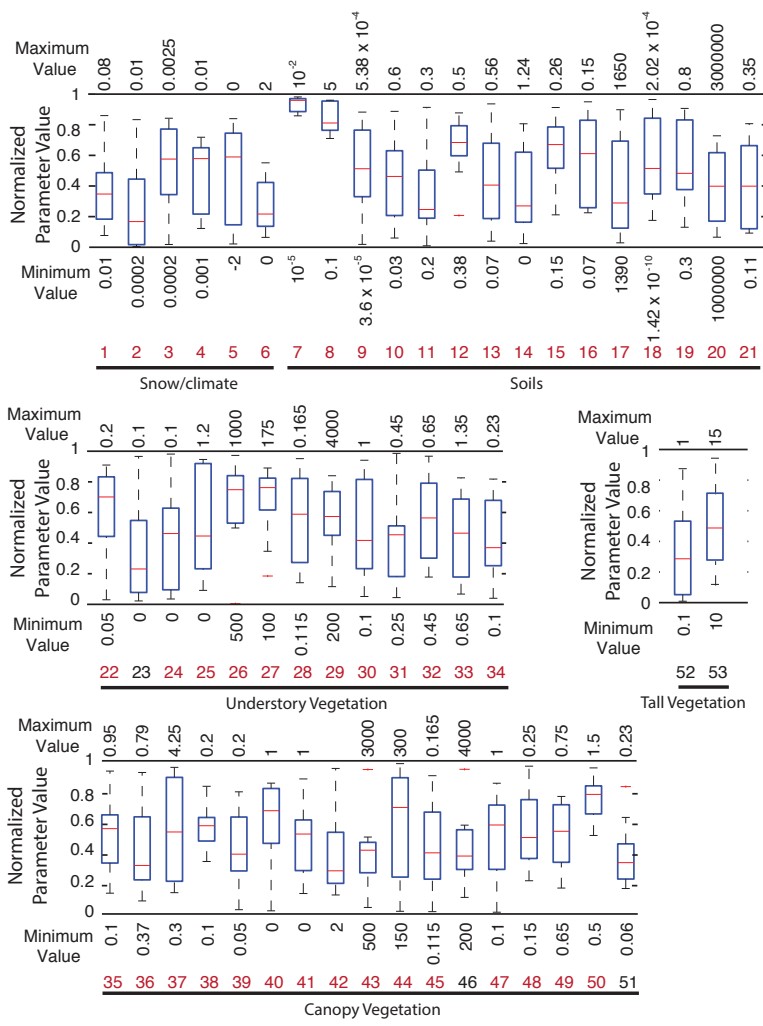

**Figure 8: Distributions of behavioural parameter values meeting framework criteria for the 53 parameters varied within the analysis. The ranges for each parameter are listed above (maximum) and below (minimum) each distribution. Distributions are normalized to values between zero and one (y-axis) to enable easier comparison (linearly scaled for all parameters except KLAT, which was $\log_{10}$ scaled). Parameters are grouped by type, with numbers referring to parameter names listed in Table 2. A two tailed Komolgov-Smirnov test was used to assess whether there was a statistically significant difference ($p < 0.1$) between the original parameter sample and the parameter values that met all framework criteria.**





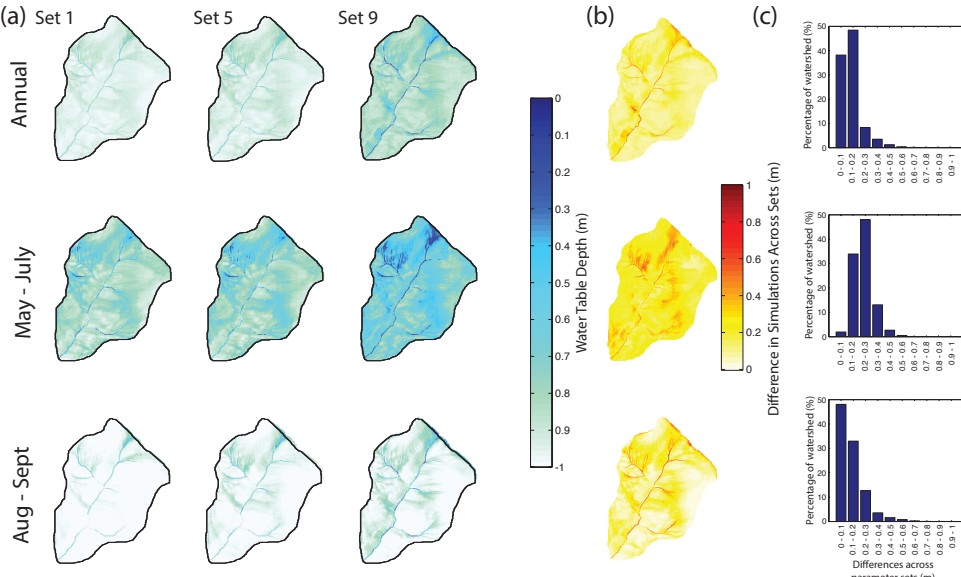

**Figure 9:** Predictions of annual, May through July, and August through September water table depths (a) in space, across the watershed, (b) shown as differences in space across nine equifinal parameter sets, and (c) with differences on a cell-by-cell basis summarized as histograms per time period. Color on maps indicates (a) the average depth across a given period between bedrock
5 (-1 m) and the surface (0 m) and (b) the largest difference between all nine predictions at each cell. Predictions are shown for three parameter sets that span the range of behaviour from drier (set 1) to average (set 5) to wetter (set 9) conditions across all nine parameter sets.



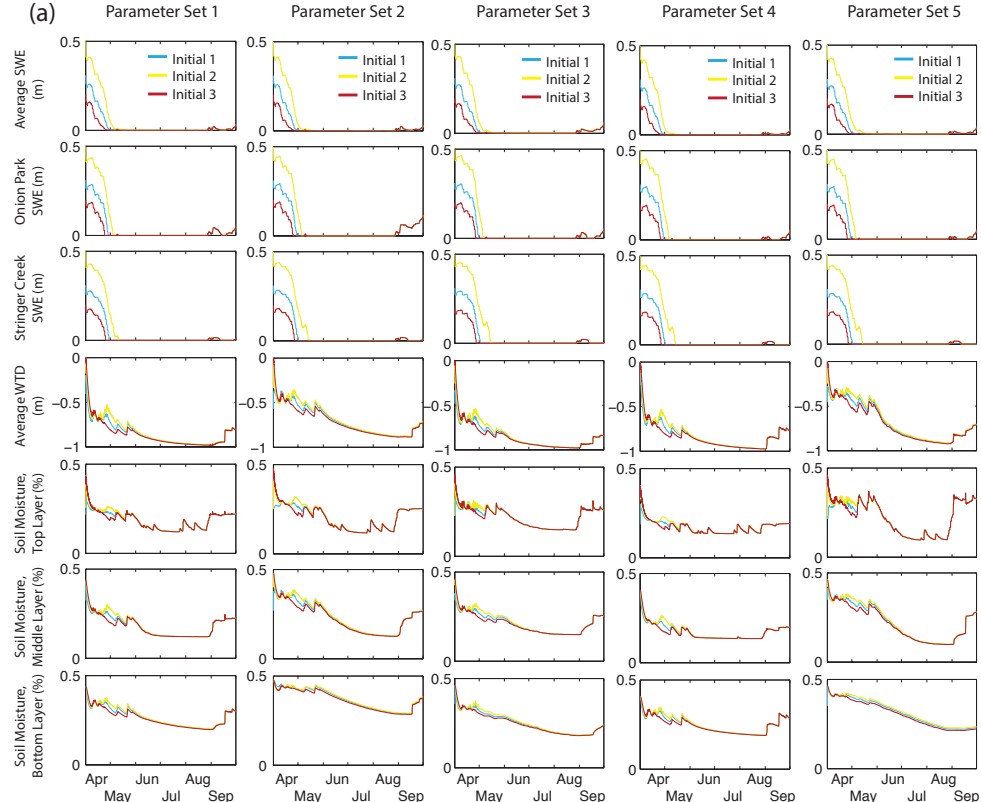

**Figure A1: Predictions of annual, May through July, and August through September water table depths (a) in space, across the watershed, (b) shown as differences in space across nine equifinal parameter sets, and (c) with differences on a cell-by-cell basis summarized as histograms per time period. Color on maps indicates (a) the average depth across a given period between bedrock (-1 m) and the surface (0 m) and (b) the largest difference between all nine predictions at each cell. Predictions are shown for three parameter sets that span the range of behaviour from drier (set 1) to average (set 5) to wetter (set 9) across all nine parameter sets.**





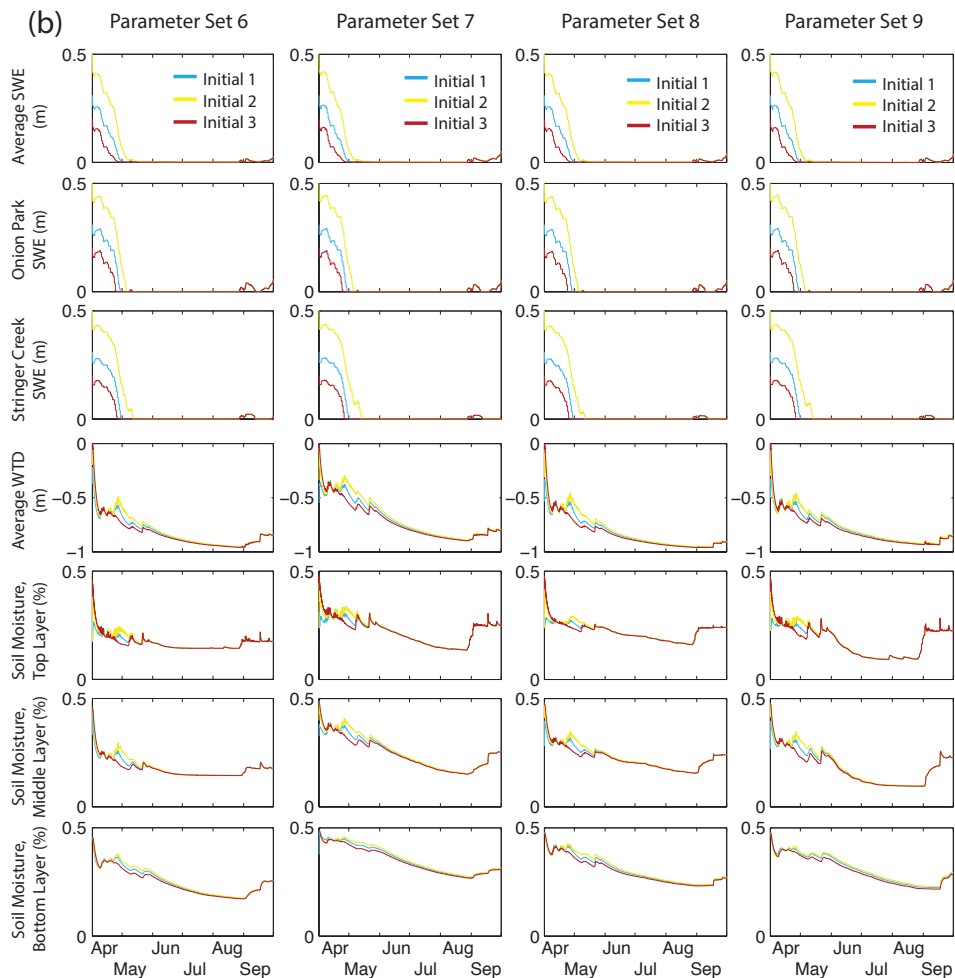

**Figure A2: Predictions of annual, May through July, and August through September water table depths (a) in space, across the watershed, (b) shown as differences in space across nine equifinal parameter sets, and (c) with differences on a cell-by-cell basis summarized as histograms per time period. Color on maps indicates (a) the average depth across a given period between bedrock (-1 m) and the surface (0 m) and (b) the largest difference between all nine predictions at each cell. Predictions are shown for three parameter sets that span the range of behaviour from drier (set 1) to average (set 5) to wetter (set 9) conditions across all nine parameter sets.**


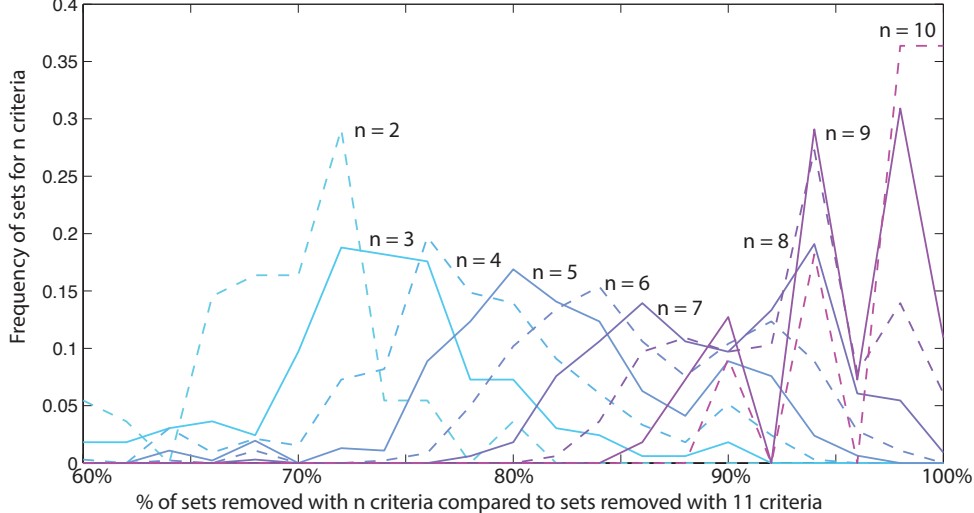

**Figure B1: This figure compares the number of nonbehavioural sets removed by subsets of different metrics benchmarked by the number of nonbehavioural sets removed by all eleven criteria. Distributions for a given subset of metrics (n = 2 to n = 10) are shown for all possible combinations of metrics.**





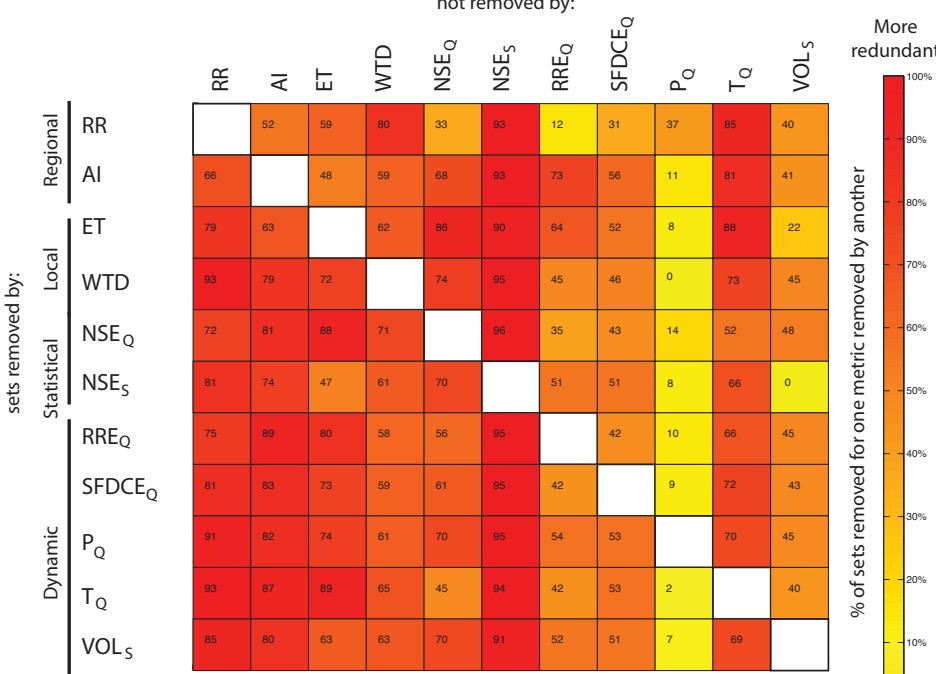

**Figure B2: Rendundancy across different metrics. The percentage of nonbehavioural sets removed by criteria applied to one metric (x-axis) that is not removed by another (y-axis).**





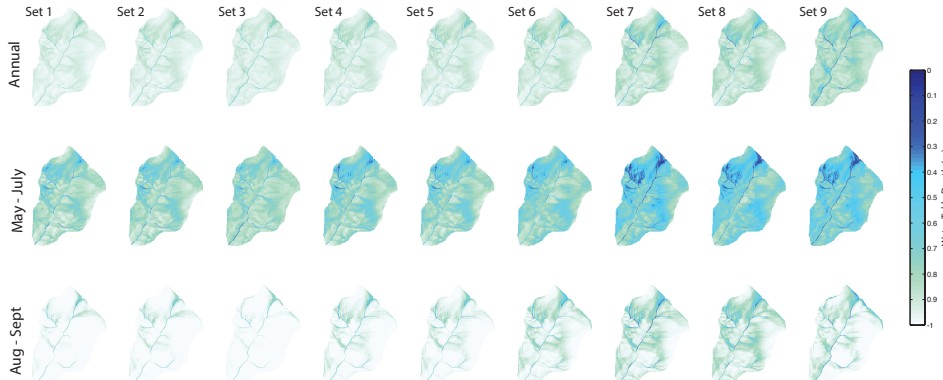

**Figure B3: Predictions of annual, May through July, and August through September water table depths. Color indicates average depth across a given period between bedrock (-1 m) and the surface (0 m). Columns display predictions across nine parameter sets identified by the framework applied to Stringer Creek hydrology.**



**Table 1: Parameter types, names, numbers (with reference to Figure 8) and ranges for all 53 parameters included in the analysis. Sources for ranges are detailed in Kelleher et al. (2015).**

| # | Name | Minimum | Maximum | # | Name | Minimum | Maximum |
|---|------|---------|---------|---|------|---------|---------|
| *Snow/climate* | | | | *Understory Vegetation* | | | |
| 1 | Snow water capacity | 0.01 | 0.08 | 27 | Minimum resistance | 100 | 175 |
| 2 | Rain LAI multiplier | 0.0002 | 0.001 | 28 | Moisture threshold | 0.115 | 0.165 |
| 3 | Snow LAI multiplier | 0.0002 | 0.0025 | 29 | Vapor pressure deficit | 200 | 4000 |
| 4 | Minimum intercepted snow | 0.001 | 0.01 | 30 | Rpc | 0.1 | 1 |
| 5 | Snow threshold | 0 | 2 | 31 | Root fraction, layer 1 | 0.25 | 0.45 |
| 6 | Rain threshold | −2 | 0 | 32 | Root fraction, layer 2 | 0.45 | 0.65 |
| *Soils* | | | | 33 | Monthly LAI | 0.65 | 1.35 |
| 7 | Lateral conductivity (KLAT) | $1\times10^{-5}$ | $1\times10^{-2}$ | 34 | Monthly albedo | 0.1 | 0.23 |
| 8 | Exponential decrease in KLAT with depth | 0.5 | 5 | *Overstory - Trees* | | | |
| | | | | 35 | Fractional coverage | 0.287 | 0.575 |
| 9 | Maximum infiltration | $3.6\times10^{-5}$ | $5.38\times10^{-4}$ | 36 | Trunk space | 0.37 | 0.79 |
| 10 | Capillary drive | 0.03 | 0.6 | 37 | Aerodynamic attenuation | 0.3 | 4.25 |
| 11 | Surface albedo | 0.2 | 0.3 | 38 | Radiation attenuation | 0.1 | 0.2 |
| 12 | Porosity | 0.38 | 0.47 | 39 | Maximum snow interception capacity | 0.05 | 0.2 |
| 13 | Pore size distribution | 0.07 | 0.559 | | | | |
| 14 | Bubbling pressure | 0 | 1.24 | 40 | Mass release drip ratio | 0 | 1 |
| 15 | Field capacity | 0.15 | 0.25 | 41 | Snow interception efficiency | 0 | 1 |
| 16 | Wilting point | 0.07 | 0.15 | 42 | Height | 2 | 7.5 |
| 17 | Bulk density | 1390 | 1650 | 43 | Maximum resistance | 500 | 3000 |
| 18 | Vertical conductivity | $1.42\times10^{-10}$ | $2.02\times10^{-4}$ | 44 | Minimum resistance | 150 | 300 |
| 19 | Thermal conductivity | 0.3 | 0.8 | 45 | Moisture threshold | 0.115 | 0.165 |
| 20 | Thermal capacity | $1\times10^{6}$ | $3\times10^{6}$ | 46 | Vapor pressure deficit | 200 | 4000 |
| 21 | Mannings n | 0.11 | 0.35 | 47 | Rpc | 0.1 | 1 |
| *Understory Vegetation* | | | | 48 | Root fraction, layer 1 | 0.65 | 0.15 |
| 22 | Maximum snow interception capacity | 0.05 | 0.2 | 49 | Root fraction, layer 2 | 0.75 | 0.25 |
| | | | | 50 | Monthly LAI | 0.5 | 1.5 |
| 23 | Mass release drip ratio | 0 | 1 | 51 | Monthly albedo | 0.06 | 0.23 |
| 24 | Snow interception efficiency | 0 | 1 | *Overstory - Tall trees* | | | |
| 25 | Height | 0 | 1.2 | 52 | Fractional coverage | 0.322 | 0.594 |
| 26 | Maximum resistance | 500 | 1000 | 53 | Height | 10 | 14.7 |





**Table 2: Signatures and error metrics used in the analysis. The table contains abbreviations for each signature/error metric, units, the general equation used for calculation, and the constraints chosen to separate behavioural and nonbehavioural runs. Abbreviations in equations refer to time step $t$, total number of time steps $m$, observed values $O$, and simulated value $S$.**

| | | Signature/ Metric | Abbrev. | Data | Units | Equation | Constraints |
|---|---|---|---|---|---|---|---|
| Signatures | Regional | Runoff Ratio | RR | Q, P | [-] | $\sum_{t=1}^{m} Q_t / \sum_{t=1}^{m} P_t$ | $0.2 < RR < 0.7$ |
| | | Aridity Index | AI | PE, P | [-] | $\sum_{t=1}^{m} PE_t / \sum_{t=1}^{m} P_t$ | $0.33 < AI < 1.206$ |
| | Local | Annual Evapotranspiration | ET | ET | [mm] | $\sum_{t=1}^{m} ET_t$ | $300mm < ET < 650\ mm$ |
| | | Average Annual Water Table Depth | WT | WT | [m] | $\frac{1}{m}\sum_{t=1}^{m} WT_t$ | $WT < 0.5m$ |
| Error Metrics | Statistical | Nash Sutcliffe Efficiency Coefficient | NSE$_Q$ | Q | [-] | $1 - \sum_{t=1}^{m}(O_t - S_t)^2 / \sum_{t=1}^{m}\left(O_t - \frac{1}{m}\sum_{t=1}^{m} O_t\right)^2$ | $NSE_Q > 0.6$ |
| | | | NSE$_S$ | SWE | [-] | | $NSE_S > 0.8$ |
| | Dynamic | Runoff Ratio Error | RR$_E$ | Q | [%] | $100 \cdot \left|\frac{\sum_{t=1}^{m} Q_{t,s}}{\sum_{t=1}^{m} P_{t,s}} - \frac{\sum_{t=1}^{m} Q_{t,o}}{\sum_{t=1}^{m} P_{t,o}}\right| \cdot \left|\frac{\sum_{t=1}^{m} P_{t,o}}{\sum_{t=1}^{m} Q_{t,o}}\right|$ | $|RR_E| < 20\%$ |
| | | Error in the Slope of the Flow Duration Curve | SFDC$_E$ | Q | [%] | $100 \cdot \left|\frac{Q_{10\%,S}-Q_{30\%,S}}{30-10} - \frac{Q_{10\%,O}-Q_{30\%,O}}{30-10}\right| \cdot \left|\frac{30-10}{Q_{10\%,O}-Q_{30\%,O}}\right|$ | $|SFDC_E| < 30\%$ |
| | | Error in Peak Q Magnitude | P$_Q$ | Q | [%] | $100 \cdot \left|P_{Q,s} - P_{Q,o}\right|\big/ P_{Q,o}$ | $|P_Q| < 35\%$ |
| | | Error in Peak Q Timing | T$_Q$ | Q | [days] | $100 \cdot \left|T_{Q,s} - T_{Q,o}\right|\big/ T_{Q,o}$ | $|P_T| < 12\ days$ |
| | | Error in SWE Volume | VOL$_S$ | SWE | [%] | $100 \cdot \left(\int_{t=1}^{m} S(t)dt - \int_{t=1}^{m} O(t)dt\right)\big/ \int_{t=1}^{m} O(t)dt$ | $|VOL_S| < 20$ |





**Table B1: Error metrics for the nine behavioural sets. Metric abbreviations correspond to metrics shown in Figures 3 and 4.**

| | | LTC | | | | | MSC | | | | | LSC | | | | |
|---|---|---|---|---|---|---|---|---|---|---|---|---|---|---|---|---|
| | | $NSE_Q$ | $RRE_Q$ | $SFDCE_Q$ | $P_Q$ | $T_Q$ | $NSE_Q$ | $RRE_Q$ | $SFDCE_Q$ | $P_Q$ | $T_Q$ | $NSE_Q$ | $RRE_Q$ | $SFDCE_Q$ | $P_Q$ | $T_Q$ |
| | | [-] | [%] | [%] | [%] | [days] | [-] | [%] | [%] | [%] | [days] | [-] | [%] | [%] | [%] | [days] |
| Water Year 2008 | Run 1 | 0.80 | 14.36 | 32.20 | -14.8 | -12.25 | 0.54 | 31.32 | 2.57 | -16.35 | 7.50 | 0.68 | -3.02 | 8.19 | -29.02 | 7.50 |
| | Run 2 | 0.83 | 1.87 | 26.19 | -19.3 | -13.00 | 0.60 | 13.53 | -5.19 | 5.73 | 7.50 | 0.70 | -16.12 | -2.43 | -22.46 | 7.50 |
| | Run 3 | 0.80 | 34.63 | 17.85 | -10.7 | -11.50 | 0.56 | 44.15 | 0.95 | -3.52 | 6.63 | 0.72 | 12.42 | -6.20 | -22.45 | 6.63 |
| | Run 4 | 0.63 | -7.38 | -0.19 | -159.7 | 8.75 | 0.47 | 25.17 | -7.88 | 0.54 | 8.75 | 0.63 | -7.38 | -1.60 | -21.84 | 8.75 |
| | Run 5 | 0.76 | 3.76 | -0.20 | -25.6 | -11.63 | 0.72 | 29.12 | -13.96 | -9.62 | -11.63 | 0.76 | 3.76 | -25.56 | -26.61 | -11.63 |
| | Run 6 | 0.72 | 0.87 | -7.65 | -22.1 | 7.75 | 0.56 | 36.60 | -11.54 | 6.94 | 4.63 | 0.72 | 0.87 | -7.65 | -23.49 | 7.75 |
| | Run 7 | 0.79 | -4.48 | -18.41 | -21.2 | 7.75 | 0.74 | 29.55 | -18.67 | -8.34 | -12.00 | 0.79 | -4.48 | -18.41 | -29.69 | 7.75 |
| | Run 8 | 0.74 | -6.86 | -3.72 | -20.3 | 9.13 | 0.63 | 26.97 | -7.80 | -3.71 | 8.63 | 0.74 | -6.86 | -3.72 | -28.19 | 9.13 |
| | Run 9 | 0.78 | -14.41 | 3.47 | -19.7 | 4.63 | 0.67 | 16.90 | 3.61 | 18.25 | 4.63 | 0.78 | -14.41 | 3.47 | -17.41 | 4.63 |
| | Minimum | 0.63 | -14.41 | -18.41 | -159.7 | -13.00 | 0.47 | 13.53 | -18.67 | -16.35 | -12.00 | 0.63 | -16.12 | -25.56 | -29.69 | -11.63 |
| | Average | 0.76 | 2.48 | 5.50 | -34.8 | -1.15 | 0.61 | 28.15 | -6.43 | -1.12 | 2.74 | 0.72 | -3.91 | -5.99 | -24.58 | 5.33 |
| | Maximum | 0.83 | 34.63 | 32.20 | -10.7 | 9.13 | 0.74 | 44.15 | 3.61 | 18.25 | 8.75 | 0.79 | 12.42 | 8.19 | -17.41 | 9.13 |
| Water Year 2007 | Run 1 | 0.80 | 23.95 | -0.55 | -1.18 | -0.50 | 0.60 | 42.90 | -33.27 | 17.89 | 1.88 | 0.77 | 9.28 | -22.17 | -2.34 | 1.88 |
| | Run 2 | 0.91 | 10.79 | 13.06 | -21.14 | -1.50 | 0.69 | 25.21 | -30.73 | -3.69 | 4.63 | 0.76 | -3.92 | -17.57 | -19.90 | 8.38 |
| | Run 3 | 0.64 | 48.94 | -6.01 | -5.04 | -4.13 | 0.58 | 59.92 | -35.76 | 14.30 | 6.38 | 0.78 | 34.54 | -31.08 | -8.78 | -1.75 |
| | Run 4 | 0.85 | 18.21 | -27.78 | -26.55 | -1.00 | 0.61 | 34.31 | -64.39 | -19.75 | 9.00 | 0.65 | 3.32 | -62.07 | -36.58 | 9.00 |
| | Run 5 | 0.78 | 48.37 | -9.65 | -18.38 | -18.00 | 0.78 | 54.61 | -35.83 | -6.32 | 2.00 | 0.82 | 35.42 | -30.08 | -26.02 | 0.38 |
| | Run 6 | 0.75 | 30.18 | 14.63 | 8.49 | -2.38 | 0.51 | 49.05 | -28.69 | 28.23 | 2.38 | 0.75 | 14.29 | -15.48 | -1.14 | 0.25 |
| | Run 7 | 0.80 | 24.85 | 11.06 | -5.44 | -3.50 | 0.71 | 39.99 | -14.98 | 7.89 | -3.00 | 0.82 | 7.45 | 0.06 | -13.26 | 1.38 |
| | Run 8 | 0.75 | 21.94 | 39.75 | 5.14 | 0.50 | 0.54 | 38.43 | -4.29 | 32.23 | 3.75 | 0.75 | 5.48 | 14.82 | 5.35 | 3.75 |
| | Run 9 | 0.83 | 11.05 | 41.99 | 6.45 | 0.88 | 0.61 | 25.69 | -10.32 | 22.52 | 2.63 | 0.75 | -4.33 | 9.20 | 2.95 | 2.63 |
| | Minimum | 0.64 | 10.79 | -27.78 | -26.55 | -18.00 | 0.51 | 25.21 | -64.39 | -19.75 | -3.00 | 0.65 | -4.33 | -62.07 | -36.58 | -1.75 |
| | Average | 0.79 | 26.48 | 8.50 | -6.40 | -3.29 | 0.63 | 41.12 | -28.69 | 10.37 | 3.29 | 0.76 | 11.28 | -17.15 | -11.08 | 2.88 |
| | Maximum | 0.91 | 48.94 | 41.99 | 8.49 | 0.88 | 0.78 | 59.92 | -4.29 | 32.23 | 9.00 | 0.82 | 35.42 | 14.82 | 5.35 | 9.00 |