# Peer review of "Characterizing and reducing equifinality by constraining a distributed catchment model with regional signatures, local observations, and process understanding"

_Hydrology and Earth System Sciences, 2016_

## Referee Comment (RC1) · M. Hrachowitz (Referee) · 4 Jan 2017

The manuscript "Characterizing and reducing equifinality by constraining a distributed catchment model with regional signatures, local observations, and process understanding" by Kelleher et al. addresses a topic that is of critical importance for hydrological modelling applications: the respective value of multiple, different data sources and model evaluation metrics to identify meaningful parameters sets within limited uncertainty. The experiment is well-designed and described. I would be glad to see this contribution eventually be published and there are only two, relatively minor concerns I

would like to encourage the authors to address to strengthen the manuscript:

(1) The manuscript would strongly benefit from being proof-read with a bit more care. The present version comes across as a bit sloppy with e.g. wrong figure numberings, wrong or missing references to figures, wrong references to or missing appendices and relatively imprecise figure captions.

(2) Being a highly important and ubiquitous topic in environmental systems models, I was quite surprised that the link to related techniques dealing with ill-posed inverse problems commonly applied in other fields is completely missing. This includes for example regularization which is quite a standard technique to reduce parameter equifinality in e.g. ground water models. It is just that it has not yet found, for whatever reason, its way into mainstream surface hydrology. Linked to that is the complete lack of this manuscript to refer to distributed model frameworks that make use of regularization (although they often refer to it as regionalization, which is the same thing, really), e.g. the work of Luis Samaniego's group on the mhM model (e.g. Samaniego et al., 2010; Kumar et al., 2013)

Detailed comments:

(3) P.1,l.11-12: distributed models do not necessarily require more parameters, if only the input is distributed (e.g. Ajami et al., 2004; Das et al., 2008; Kling and Gupta, 2009; Fenicia et al., 2008; Euser et al., 2015). I would suggest to make this distinction clear somewhere in the introduction section.

(4) P.1,l.18: I find the term "certainty" quite problematic. Given all the different sources of uncertainty, how can we ever think of a parameter as being "certain". What would that mean in that context? "Certain" with respect to what? Reality? Uncertain observations?

(5) P.1,l.27ff: it is not quite clear what the authors define as distributed model in this study. Note that both, physical models but also conceptual models (for example done

with mhM) can be applied in distributed ways. Please provide a more precise definition.

(6) P.2,l.2: maybe rephrase to "Distributed models should represent the. . . ."

(7) P.2,l.10: why would distributed models *require* a single parameter set? This statement goes against much we know about the uncertainties involved in environmental systems models

(8) P.2,l.14-15: not only valid for distributed models but for *any* hydrological model

(9) P.2,l.19-21: not untrue, but the contribution of the developers of the mhM framework, who went to great lengths to exploit the value of regionalization/regularization, should not go unnoticed here (see above)

(10) P.2,l.27: I would suggest to formulate this in a more general way by emphasizing the degree of information that is used on the prior distributions

(11) P.3,l.3: should read as ". . .to justify the selection. . ."

(12) P.4,l.18,figure1: where do all the tall trees hide?

(13) P.5,l.2: what does "tandem" mean?

(14) P.5,l.7,table 1: add units in table 1; what is referred to as "ranges"? is it the uninformed prior parameter distributions? Please clarify.

(15) P.5,l.24: please be more specific here – what does shading include? Does it combine shading due to aspect and shading due to topographic features (e.g. mountain ridge) obstructing sunlight to reach certain locations?

(16) P.5,l.26: please be more specific here – what are "allowable" ranges?

(17) P.5,l.30ff: not entirely clear – which period was the model calibrated for? The entire 01/10/2006-01/10/2008 period? If so, I was wondering if it was not more instructive to retain some months for post-calibration evaluation (i.e. "validation") to see the effect of signatures etc. under these conditions as well.

(18) P.8,l.27,table 2: which runoff ratio was used? Over the entire period? Annual runoff ratios? Or seasonal runoff ratios? They carry different information content, and from earlier work we found that in particular the seasonal runoff ratios carry considerable information. This may be worth looking into.

(19) P.8,l.28ff,table 2: PET is constrained with regional data, but how were these regional PETs estimated? Are they more reliable than the penmen-monteith derived estimates here? if so, why? If not, what is the point of using them. I think this point warrants some discussion.

(20) P.9,l.15,table 2: does this refer to the maximum flow of each year? In many cases in particular the extreme events are subject to unproportionally high observation errors. Thus, do you not run into the risk of forcing the model to reproduce a value that is relatively likely to be considerably off from reality (cf. epistemic errors!!)? I also think that this point needs some more reflection.

(21) P.9,l.16,table 2: the equation for the error in peak timing (again: does this apply for the timing in each individual year?) seems to involve some magic – I could not figure out how a ratio of days over days would result in days. In addition, I am not sure if this ratio makes sense, as we talk about the day of the year, at least as I understand it. Would be probably more meaningful to just minimize or constrain the absolute errors, i.e. abs(Tq,s-Tq,o)

(22) P.10,l.8ff,figure 4: this is not entirely clear. Do the grey and coloured bars show the parameter sets that are retained or those that are discarded? The text and figure captions seem to somehow contradict each other. Please check.

(23) P.10,l.10-11: what about the effect of observational uncertainty in precipitation (in particular snow accumulation!!) and peak runoff - see comment (20)??

(24) P.11,l.11,figure 6: I am struggling with this figure. Caption seems to describe an entirely different figure. Where is the black dotted line (the one I see merely seems to

separate the two hydrological years)? Where are subplots (a) and (b)?? what is the light blue line (observation?)? please clarify.

(25) P.12, l.3, figure 8(?): figure number seems to be wrong. Is this not figure 7? if not, where is figure 7? Please indicate years on x-axes. What are the grey shaded areas? Are these model ensembles or uncertainty ranges? If the latter, how were they constructed and, in particular, where do the gaps come from??

(26) P.12,l.4: what is meant by "relative timescales"?

(27) P.12,l.17: I cannot find Appendix D (and C not either, for that matter).

(28) P.12,l.23: please reconsider the use of the term "parameter certainty" (see above)

(29) P.12,l.30: what is meant by "original" distribution? Is it the uninformed prior distribution (i.e. table 1)? If so, please call it also like that. In addition, although I see and understand the intention here, I am wondering how much this also depends on the choice of the parameter range in the prior distribution. If a narrow prior range was chosen, the normalized values shown here may be less constrained than if a wider prior range was chosen. in addition, are the individual parameter sets likelihood weighted to give more wight to better solutions? if not, why not? Any thoughts on this?

(30) P.13,l.6: where are these predictions included? Figure b3? Please specify.

(31) P.13, l.13, figure 9: the figure and captions are a bit confusing. What is meant by set 1/5/9? And how is set 1 drier than set 9? Please clarify.

(32) P.18,l.6: I think the wrong reference is provided in the reference list. The paper you are referring to is Euser et al. (2015, HP), but in the reference list Euser et al. (2013, HESS) is provided.

(33) P.18, l.9: some references to mhM would fit in nicely here.

Best regards, Markus Hrachowitz

References:

Ajami, N. K., Gupta, H., Wagener, T., & Sorooshian, S. (2004). Calibration of a semi-distributed hydrologic model for streamflow estimation along a river system. Journal of Hydrology, 298(1), 112-135.

Das, T., Bárdossy, A., Zehe, E., & He, Y. (2008). Comparison of conceptual model performance using different representations of spatial variability. Journal of Hydrology, 356(1), 106-118.

Euser, T., Hrachowitz, M., Winsemius, H. C., & Savenije, H. H. (2015). The effect of forcing and landscape distribution on performance and consistency of model structures. Hydrological Processes, 29(17), 3727-3743.

Fenicia, F., Savenije, H. H., Matgen, P., & Pfister, L. (2008). Understanding catchment behavior through stepwise model concept improvement. Water Resources Research, 44(1).

Kling, H., & Gupta, H. (2009). On the development of regionalization relationships for lumped watershed models: The impact of ignoring sub-basin scale variability. Journal of Hydrology, 373(3), 337-351.

Kumar, R., Livneh, B., & Samaniego, L. (2013). Toward computationally efficient large‐scale hydrologic predictions with a multiscale regionalization scheme. Water Resources Research, 49(9), 5700-5714.

Kumar, R., Samaniego, L., & Attinger, S. (2013). Implications of distributed hydrologic model parameterization on water fluxes at multiple scales and locations. Water Resources Research, 49(1), 360-379.

Samaniego, L., Kumar, R., & Attinger, S. (2010). Multiscale parameter regionalization of a grid‐based hydrologic model at the mesoscale. Water Resources Research, 46(5).

---

## Referee Comment (RC2) · U. Ehret (Referee) · 18 Jan 2017

**Review of Manuscript**

**'Characterizing and reducing equifinality by constraining a distributed catchment model with regional signatures, local observations, and process understanding'**

**by C. Kelleher et al.**

Dear Editor, dear Authors,

I have reviewed the aforementioned work (version 2 of the manuscript). My conclusions and comments are as follows:

**1. Scope**

The article is within the scope of HESS.

**2. Summary**

The authors present and apply a framework for calibration of distributed hydrological models by applying a hierarchical set of parameter constraints and error metrics to accept or reject randomly generated parameter sets. The constraints and error metrics are distinguished by i) range of applicability (from regional to local), ii) 'softness' (from local observations to heuristically formulated local expert knowledge) and iii) the evaluated characteristic (evaluation of non-dynamical to dynamical aspects).

The framework is presented at the example of the physically based, distributed DHSVM model applied to the 5.5 km² Stringer Creek catchment, whose hydrological behavior is dominated by seasonal snow accumulation and snow melt.

Using 10.000 randomly drawn model parameter sets, 10 signatures and error metrics are applied individually, in groups and in hierarchical combination to identify behavioral parameter sets from the initial set. The resulting subsets are then discussed with respect to the equifinality reduced (i.e. by how much the initial parameter ranges were narrowed).

The authors show that by jointly applying all criteria considerably narrows the behavioral parameter sets (here: to nine). However, these still show large differences, specifically with respect to catchment groundwater table (values and spatial patterns).

From the analysis, the authors conclude that i) a multi-criteria approach to identification of behavioral parameter sets is superior to single-criteria approaches, ii) dynamic constraints to be more effective than non-dynamical ones, and iii) that despite substantial narrowing of the parameter space still large differences among the surviving parameter sets remain, especially with respect to spatial patterns of hydrologic states.

**3. Overall ranking**

The work is ranked 'Major revision'.

**4. Evaluation**

**Major points**

I like the work presented in this paper, especially the strong argument towards using multiple, hard and soft sources of information to identify behavioral model parameter sets, and the thorough literature review. However, despite the fact that the authors' focus in this paper is to present the concept, with the choice of the catchment and time series used, they have clearly missed some very good opportunities to make their results more general and interpretable. More specific:

- Judging from the presented time series of discharge and snow water equivalent, the catchments' hydrological function is very simple (one major discharge event during snowmelt, rainfall-runoff events are hardly playing a role). Arguably, a very simple conceptual hydrological model could reproduce this behavior at least as well as the applied model with respect to all discharge-related signatures and metrics, but with much less parameters.
  - So why choose this very simple-behaving catchment if the goal is demonstrate the usefulness of a targeted constraining approach for a distributed, physically based model? This way, the model stays well below its potential, and this also means a lot of opportunities for more targeted model parameter evaluation are missed.
  - Furthermore, the many degrees of freedom in your model inevitably lead to problems of equifinality, which would not exist in a simpler model appropriate for the simple catchment. So why not choose a more complex catchment, which requires a distributed model?
- Along the same lines: Why was a catchment selected with such little available observations as 'hard truth'? Why not choose one with a network of observed groundwater tables, ET, nested discharge observations, spatially distributed information on soil type and soil depth etc.? In fact, the model elements are all set to the same soil depth and soil type, which makes it much less distributed as it could be. Placing the study in a better equipped catchment would offer the opportunity to fill the very nice framework with many more signatures and metrics, especially those evaluating spatial patterns. Furthermore, this would have opened the opportunity to compare the value of constraints formulated as aggregated/heuristic expert knowledge to 'hard' constraints based on observations. This is a clear miss.
- Along the same lines: All evaluations are done for a single year, and for the calibration period. This way it is impossible to judge
  - to which degree the remaining behavioral parameter sets are dependent on the chosen calibration period, and
  - whether the behavioral parameter sets found in calibration are still behavioral during a different, validation time. This is a clear miss.
- The authors advocate a multi-criteria approach to identify behavioral parameter sets. However, looking at the criteria in Table 2, two questions arise
  - Parameterization of ET plays an important role in the model (parameters 22 to 53 in Table 1). However, ET is only used as a very weak constraint (300-650 mm/a). Why not also evaluate the model with respect to ET error, ET timing, ET peaks etc.? From the text, my understanding is that ET estimates from local observations exist. This would be another important and independent criterion.
  - I do not understand the usage of AI: From my understanding of the text, both PE and P are from observations, so AI is independent of the model. So how can this criterion be used as a signature for model evaluation?

**Minor points**
- From my experience, the main control on parameter equifinality is model structural choice. The authors discuss this important issue briefly in the conclusions. I encourage them to discuss this aspect in more detail, although I am well aware that model structural choice is not the topic of the paper. However, it can offer an avenue of progress to reduce the still-high equifinality of the final behavioral set of parameter sets.
- P8/l29: RR instead of PET?
- P12/L17: Where is Appendix D?
- P13/L3: Why not also compare simulations to well observations? (see also my major comments)
- P13/L7: Appendix B3 instead of B only?
- P14/L22-23: … suggest that evaluating certain types of internal behavior by point observations…?

- P21/L14: Figure B3 instead of Fig 9?
- Fig 1:
    - A) The location of the label 'Tenderfoot creek' is misleading
    - A) is the scale really [km]?
    - C) add a legend (which gauge is which)
- Fig 4, caption: remaining instead of removed?
- Fig 6, caption: Where are the black dotted lines? Where is (a) and (b)?
- Fig 7: Please add year indicators (2007, 2008), a legend (which gauge is which) and for clarity add in the caption that these are plots for the final subset of 9 parameter sets
- Fig 8, caption: For clarity please add in the caption that these are plots for the final subset of 9 parameter sets.

Yours sincerely,

Uwe Ehret

---

## Author Comment (AC1) · 16 Mar 2017

**Dear Dr. Hrachowitz,**
**We thank you for your suggested revisions. In line with your comments, we have revised and checked all figure captions, references, and numbering of figures, tables, and appendices. Our introduction and discussion sections now include references to examples of regularization in watershed modeling, an oversight on our part. We have also addressed many of your additional minor comments, which were helpful for clarifying the direction of the manuscript. We have responded to each of your comments below (in red text), and noted changes we have made in response.**

**Many thanks,**
**Christa Kelleher and Colleagues**

The manuscript "Characterizing and reducing equifinality by constraining a distributed catchment model with regional signatures, local observations, and process understanding" by Kelleher et al. addresses a topic that is of critical importance for hydrological modelling applications: the respective value of multiple, different data sources and model evaluation metrics to identify meaningful parameters sets within limited uncertainty. The experiment is well-designed and described. I would be glad to see this contribution eventually be published and there are only two, relatively minor concerns I would like to encourage the authors to address to strengthen the manuscript:

(1) The manuscript would strongly benefit from being proof-read with a bit more care. The present version comes across as a bit sloppy with e.g. wrong figure numberings, wrong or missing references to figures, wrong references to or missing appendices and relatively imprecise figure captions.

In line with these suggestions, we have revised captions and checked mentions of references, figures, and appendices.

(2) Being a highly important and ubiquitous topic in environmental systems models, I was quite surprised that the link to related techniques dealing with ill-posed inverse problems commonly applied in other fields is completely missing. This includes for example regularization which is quite a standard technique to reduce parameter equifinality in e.g. ground water models. It is just that it has not yet found, for whatever reason, its way into mainstream surface hydrology. Linked to that is the complete lack of this manuscript to refer to distributed model frameworks that make use of regularization (although they often refer to it as regionalization, which is the same thing, really), e.g. the work of Luis Samaniego's group on the mhM model (e.g. Samaniego et al., 2010; Kumar et al., 2013)

We appreciate the recommendation to include this subset of literature in our revised manuscript. We have made changes to the introduction and discussion to address this oversight.

We have added the following to the introduction (page 3): "One noted exception is the work that has been done to parameterize models via regularization (Hundecha and Bardossy, 2004; Hundecha et al., 2008; Samaniego et al., 2010; Rakovec et al., 2016). Regularization creates global functional relationships describing transfer functions that link model parameters and

catchment characteristics (e.g., Samaniego et al., 2010; Kumar et al., 2013). Regularization has the ability to limit parameter equifinality in part because the number of parameters used to describe global functional relationships is far fewer than the number of possible total model parameters using cell-by-cell parameterization."

Detailed comments:
(3) P.1,l.11-12: distributed models do not necessarily require more parameters, if only the input is distributed (e.g. Ajami et al., 2004; Das et al., 2008; Kling and Gupta, 2009; Fenicia et al., 2008; Euser et al., 2015). I would suggest to make this distinction clear somewhere in the introduction section.

To address this point, we have amended the introduction to include a paragraph describing different types of distributed models. In particular, we include the statement:
"Alongside variations in the structural equations, models may take a number of different forms, including those that use only distributed inputs (e.g., Ajami et al., 2004; Das et al., 2008; Kling and Gupta, 2009; Fenicia et al., 2008; Euser et al., 2015), those that may lump together portions of a watershed with similar characteristics in a semi-distributed fashion (e.g., Leavesley et al., 1983; Flugel, 1995; Ajami et al., 2004; Das et al., 2008; Zehe et al., 2014), those that may upscale sub-grid variability to parameterize models (e.g., Samaniego et al., 2010; Kumar et al., 2013), and those that use physically-based parameter values (e.g., Wigmosta et al, 1994; Qu and Duffy, 2007) versus conceptual values or transfer functions (e.g., Samaniego et al., 2010; Kumar et al., 2013).

(4) P.1,l.18: I find the term "certainty" quite problematic. Given all the different sources of uncertainty, how can we ever think of a parameter as being "certain". What would that mean in that context? "Certain" with respect to what? Reality? Uncertain observations?

We have removed the term certainty from the abstract, and all other mentions of this terminology from the manuscript.

(5) P.1,l.27ff: it is not quite clear what the authors define as distributed model in this study. Note that both, physical models but also conceptual models (for example done with mhM) can be applied in distributed ways. Please provide a more precise definition.

We see this as a very important point. We have clarified this point by adding a paragraph to the introduction (page 2): "Alongside variations in the structural equations, models may take a number of different forms, including those that use only distributed inputs (e.g., Ajami et al., 2004; Das et al., 2008; Kling and Gupta, 2009; Fenicia et al., 2008; Euser et al., 2015), those that may lump together portions of a watershed with similar characteristics in a semi-distributed fashion (e.g., Leavesley et al., 1983; Flugel, 1995; Zehe et al., 2014), those that may upscale sub-grid variability to parameterize models (e.g., Samaniego et al., 2010; Kumar et al., 2013), and those that use physically-based parameter values (e.g., Wigmosta et al, 1994; Qu and Duffy, 2007) versus conceptual values or transfer functions (e.g., Samaniego et al., 2010; Kumar et al., 2013). These differences can be separated into two primary categories (1) the nature of parameter values (physically-based or conceptual) and (2) whether and how parameter values are distributed (cell-by-cell, grouped or lumped in some meaningful way, or undistributed). It is also

worth noting that an important feedback on these decisions is the scale of the application alongside the scale at which inputs and parameters are distributed. In this study, we focus specifically on physically-based, fully distributed (cell-by-cell basis) models applied at the small-scale watershed scale (<25 km$^2$)."

(6) P.2,l.2: maybe rephrase to "Distributed models should represent the. . .."

We have made this change.

(7) P.2,l.10: why would distributed models *require* a single parameter set? This statement goes against much we know about the uncertainties involved in environmental systems models

While we agree with you (in fact, addressing approaches that use a single parameter set is the point of this paper), the reality is this is still how many researchers use fully distributed, physically-based models. We have revised this statement to:

"Model simulations and predictions require specification of parameter set(s) or ranges; selecting these set(s) and appropriate ranges is especially challenging given that distributed models require a larger number of model parameters (~50-100 or more) and longer model run times than conceptual or lumped models."

(8) P.2,l.14-15: not only valid for distributed models but for *any* hydrological model

We agree, and have altered the text to reflect this.

(9) P.2,l.19-21: not untrue, but the contribution of the developers of the mhM framework, who went to great lengths to exploit the value of regionalization/regularization, should not go unnoticed here (see above)

We have included the following text on page 3: One noted exception is the work that has been done to parameterize models via regularization (Hundecha and Bardossy, 2004; Hundecha et al., 2008; Pohkrel et al., 2008; Samaniego et al., 2010; Rakovec et al., 2016). Regularization creates global functional relationships describing transfer functions that link model parameters and catchment characteristics (e.g., Samaniego et al., 2010; Kumar et al., 2013). As the number of parameters used to describe global functional relationships is far fewer than the number of possible total model parameters using cell-by-cell parameterization, regularization is able to limit parameter equifinality."

(10) P.2,l.27: I would suggest to formulate this in a more general way by emphasizing the degree of information that is used on the prior distributions

We have changed this to:
"Decisions regarding how to constrain ranges and distributions for priors on model parameters often depend on…"

(11) P.3,l.3: should read as ". . .to justify the selection. . ."

Noted and changed.

(12) P.4,l.18,figure1: where do all the tall trees hide?

There are a small number of tall trees, and a very large matrix of values (our simulated grid is 560 by 760 cells). We have increased the contrast of this figure to highlight tall trees, but note that the dominating pattern here are the differences between understory and canopy vegetation.

(13) P.5,l.2: what does "tandem" mean?

By 'In tandem' we meant 'at the same time' – we have changed this to 'concurrently'.

(14) P.5,l.7,table 1: add units in table 1; what is referred to as "ranges"? is it the uninformed prior parameter distributions? Please clarify.

We have added units to Table 1, and clarified our description of parameter ranges as "the minimum and maximum bounds on uninformed (uniform) prior parameter distributions".

(15) P.5,l.24: please be more specific here – what does shading include? Does it combine shading due to aspect and shading due to topographic features (e.g. mountain ridge) obstructing sunlight to reach certain locations?

We have clarified this point on page 6 with the following text: "These shading maps incorporate both the effects of topographic shading, diel variability, and monthly variability in sun position and angle."

(16) P.5,l.26: please be more specific here – what are "allowable" ranges?

This became a problem with spatial boundary conditions, where humidity and vapor pressure deficit on the coldest days (< -40 deg F) at the far edges of the model reached unrealistic conditions. We have amended the word 'allowable' to 'unrealistic'.

(17) P.5,l.30ff: not entirely clear – which period was the model calibrated for? The entire 01/10/2006-01/10/2008 period? If so, I was wondering if it was not more instructive to retain some months for post-calibration evaluation (i.e. "validation") to see the effect of signatures etc. under these conditions as well.

As stated in the text (Page 6 lines 27 – 29), we use the 2008 water year for calibration, and retain the 2007 water year for validation. We use April 1, 2006 through September 30, 2006 to initialize the model. We have added dates for clarity, and have moved this text to earlier in the paragraph. We exclude a period of initialization (Apr 1, 2006 – Sept 30, 2006) from any model analysis.

(18) P.8,l.27,table 2: which runoff ratio was used? Over the entire period? Annual runoff ratios? Or seasonal runoff ratios? They carry different information content, and from earlier work we

found that in particular the seasonal runoff ratios carry considerable information. This may be worth looking into.

We use an annual runoff ratio for the 2008 water year.  To target seasonality, we use the slope of the flow duration curve for 10[th] through 30[th] percentile flows (mainly to fit to the rising and falling limbs of the hydrograph), though a seasonal runoff ratio is an interesting suggestion for future analysis to ensure model simulations match seasonal behavior.

(19) P.8,l.28ff,table 2: PET is constrained with regional data, but how were these regional PETs estimated? Are they more reliable than the penmen-monteith derived estimates here? if so, why? If not, what is the point of using them. I think this point warrants some discussion.

The estimate of aridity used to constrain the model is an average value based on daily precipitation and daily maximum and minimum air temperature data for the period of 1950 to 2000, with potential evapotranspiration (PET) modeled via the Hargreaves method (Hargreaves et al., 1985).  Thus, this constraint is meant as a check on long-term average conditions, to ensure that the model stays within regionally realistic estimates.  We have modified the text to reflect this point:

"Constraints on AI from this dataset represent long-term average conditions for the region, and are used to broadly eliminate simulations outside of the realm of regionally realistic average estimates."

To corroborate the validity of this 'space-for-time' approach, we calculated PET following a combination method described by Oudin et al. (2005) and calculated AI from estimated PET and annual precipitation from the Stringer Creek SNOTEL location.  Values for AI ranged from 0.39 to 0.64, well within the constraints extracted from the regional estimates.

Table 1. Estimates for aridity index (AI) based on annual precipitation (Stringer Creek SNOTEL) and potential evapotranspiration calculated following Oudin et al. (2005).

| Year | PET (m) | P (m) | AI (-) |
| --- | --- | --- | --- |
| 2006 | 0.47 | 0.80 | 0.59 |
| 2007 | 0.48 | 0.75 | 0.64 |
| 2008 | 0.41 | 0.73 | 0.56 |
| 2009 | 0.43 | 0.74 | 0.57 |
| 2010 | 0.40 | 1.04 | 0.39 |
| 2011 | 0.41 | 0.86 | 0.47 |

DHSVM uses a cascading approach (from overstory to the understory) to estimate potential evapotranspiration, with differences largely based on vegetation height parametrizing aerodynamic resistance.  As indicated by Figure B2, AI does appear to have information content, but it is largely with respect to very poor model runs likely produced as the combination of the extreme vegetation height parameter combinations.

To test the effect of constraining AI on the final set of parameters, we recalculated the number of parameter sets that would meet all constraints excluding the constraint on AI. We found that this number would increase from 9 to 13 parameter sets, demonstrating that this constraint may only serve to remove a small number of sets that do not meet other constraints. We have added a figure demonstrating how the number of final parameter sets changes if only one constraint is removed to Appendix B, Figure B3, as this addresses points raised by both reviewers.

(20) P.9,l.15,table 2: does this refer to the maximum flow of each year? In many cases in particular the extreme events are subject to unproportionally high observation errors. Thus, do you not run into the risk of forcing the model to reproduce a value that is relatively likely to be considerably off from reality (cf. epistemic errors!!)? I also think that this point needs some more reflection.

This metric does refer to the maximum flow of each year. We found that without this constraint, the model consistently underestimated peak flows. We do agree with your point regarding potential for fitting to epistemic errors. This is in part why we left windows on acceptable "errors" wide (e.g., +/- 35%), so as to ensure that we do not overly constrain these single observation points in time. While we agree that overly constraining this value could lead to problems, in previous iterations of this manuscript, criticisms of our modeling approach primarily focused on whether or not our simulations emulated this peak value. Thus, we aimed to add a target error for this value, but one that was wide.

To reflect this, we have added the following text to page 10:
"Additionally, as peak flow is likely to be most influenced by epistemic errors (e.g., Westerberg and McMillan, 2015), we left bounds on peak flow to be reasonably wide, so as to avoid incorrectly constraining a potentially uncertain measurement."

(21) P.9,l.16,table 2: the equation for the error in peak timing (again: does this apply for the timing in each individual year?) seems to involve some magic – I could not figure out how a ratio of days over days would result in days. In addition, I am not sure if this ratio makes sense, as we talk about the day of the year, at least as I understand it. Would be probably more meaningful to just minimize or constrain the absolute errors, i.e. abs(Tq,s-Tq,o)

This was an error in Table 2, and has been corrected. The constraints were correctly identified.

(22) P.10,l.8ff,figure 4: this is not entirely clear. Do the grey and coloured bars show the parameter sets that are retained or those that are discarded? The text and figure captions seem to somehow contradict each other. Please check.

We note the confusion, and have corrected the captions to 'retained'.

(23) P.10,l.10-11: what about the effect of observational uncertainty in precipitation (in particular snow accumulation!!) and peak runoff - see comment (20)??

Again, we note that we are not fitting exactly to this value, but including any model simulations that are within +/- 35% of this value. We agree that there are likely to be uncertainties in these

measurements, hence our choice to leave bounds on these values to be wide. As many sets
match SWE behavior, we do not think we are overly constraining peak flow behavior on the
basis of observational uncertainty in snow accumulation.

(24) P.11,l.11,figure 6: I am struggling with this figure. Caption seems to describe an entirely
different figure. Where is the black dotted line (the one I see merely seems to separate the two
hydrological years)? Where are subplots (a) and (b)?? what is the light blue line (observation?)?
please clarify.

This figure caption has been corrected.

(25) P.12, l.3, figure 8(?): figure number seems to be wrong. Is this not figure 7? if not, where is
figure 7? Please indicate years on x-axes. What are the grey shaded areas? Are these model
ensembles or uncertainty ranges? If the latter, how were they constructed and, in particular,
where do the gaps come from??

We have corrected the text to state this is Figure 7. We have added years to the x-axis on Figure
7. The grey lines each correspond to a simulation from a different parameter set. We have
clarified the figure caption to reflect this point.

(26) P.12,l.4: what is meant by "relative timescales"?

We have amended this to "match periods" for clarity.

(27) P.12,l.17: I cannot find Appendix D (and C not either, for that matter).

This has been changed to 'Table B1'.

(28) P.12,l.23: please reconsider the use of the term "parameter certainty" (see above)

We have changed this to 'Parameter Uncertainty'.

(29) P.12,l.30: what is meant by "original" distribution? Is it the uninformed prior distribution
(i.e. table 1)? If so, please call it also like that.

This has been changed.

In addition, although I see and understand the intention here, I am wondering how much this also
depends on the choice of the parameter range in the prior distribution. If a narrow prior range
was chosen, the normalized values shown here may be less constrained than if a wider prior
range was chosen.

We agree that this depends on the width and magnitude of the uninformed priors. While this
likely has an effect, we do note that parameters where ranges appear to be more constrained do
have a variety of magnitudes and ranges. For instance, lateral conductivity (parameter 5), varies
across three orders of magnitude, exponential decrease in conductivity with depth (parameter 6),

varies across two orders of magnitude, and understory maximum stomatal resistance (parameter 26) varies across one order of magnitude. The meaning of "narrow" and "wide" ranges also has some implications for our assessment – we intentionally left all ranges relatively wide in terms of values and ranges reported in the literature. Our point with this figure is mainly to highlight that there are a few parameters where ranges are narrowed by this type of analysis, which is interesting given the number of interactions between parameters in models of this type.

In addition, are the individual parameter sets likelihood weighted to give more weight to better solutions? if not, why not? Any thoughts on this?

We did not assign likelihood weights to parameter sets, in part because the parameter sets shown in this figure meet all criteria for what we have identified to be 'behavioral' solutions. We are mixing both signatures and error metrics, as well as a number of different signatures and error metrics. Thus, while some may have better values for one objective function, they may have poorer values for another. As there was no clear or easy way to "weight" these parameter sets across these many 'behavioral' criteria, we opted to leave these values unweighted.

(30) P.13,l.6: where are these predictions included? Figure b3? Please specify.

This has been changed.

(31) P.13, l.13, figure 9: the figure and captions are a bit confusing. What is meant by set 1/5/9? And how is set 1 drier than set 9? Please clarify.

This labeling originally referred to Figure B3. We have replaced the labeling with sets 1, 2, and 3, to indicate that these predictions come from three different parameter sets, and now reference Figure B3 in the caption. We have also replaced the words wetter and drier with "lower" and "higher" water table.

(32) P.18,l.6: I think the wrong reference is provided in the reference list. The paper you are referring to is Euser et al. (2015, HP), but in the reference list Euser et al. (2013, HESS) is provided.

We have corrected this.

(33) P.18, l.9: some references to mhM would fit in nicely here.

We have included the following text on page 19 (section 5.6):
"In this vein, the choice of model structure may also offer another opportunity to reduce equifinality. In particular, the extensive body of literature on parameter regularization may offer a pathway for maintaining spatial complexity and consistency while reducing the number of free model parameters (Hundecha and Bardossy, 2004; Hundecha et al., 2008; Samaniego et al., 2010; Rakovec et al., 2016). Alternatively, there is also a body of work that treats the model framework itself as a form of uncertainty, testing different model structures as hypotheses for how a catchment may function (Clark et al., 2011; Fenicia et al., 2011; Hrachowitz et al., 2014). This approach may also provide an alternative to predicting hydrology via a model with fewer

[revised manuscript text omitted]

Zehe, E., Ehret, U., Pfister, L., Blume, T., Schröder, B., Westhoff, M., Jackisch, C., Schymanski, S. J., Weiler, M., Schulz, K., Allroggen, N., Tronicke, J., van Schaik, L., Dietrich, P., Scherer, U., Eccard, J., Wulfmeyer, V., and Kleidon, A.: HESS Opinions: From response units to

functional units: a thermodynamic reinterpretation of the HRU concept to link spatial organization and functioning of intermediate scale catchments, Hydrol. Earth Syst. Sci., 18, 4635-4655, doi:10.5194/hess-18-4635-2014, 2014.

---

## Author Comment (AC2) · 16 Mar 2017

**Dear Dr. Ehret,**
**Thank you for your thoughtful critique of our manuscript. We have addressed each of your comments below (in red text). In particular, your comments have helped to clarify the intent of our manuscript, as well as ensuring that this intent is clearly stated in justification of our approach. These clarifications were focused on the introductory and discussion portions of the text.**

**Christa Kelleher and Colleagues**

Dear Editor, dear Authors,

I have reviewed the aforementioned work (version 2 of the manuscript). My conclusions and comments are as follows:

1. Scope

The article is within the scope of HESS.

2. Summary

The authors present and apply a framework for calibration of distributed hydrological models by applying a hierarchical set of parameter constraints and error metrics to accept or reject randomly generated parameter sets. The constraints and error metrics are distinguished by i) range of applicability (from regional to local), ii) 'softness' (from local observations to heuristically formulated local expert knowledge) and iii) the evaluated characteristic (evaluation of non-dynamical to dynamical aspects).

The framework is presented at the example of the physically based, distributed DHSVM model applied to the 5.5 km² Stringer Creek catchment, whose hydrological behavior is dominated by seasonal snow accumulation and snow melt.

Using 10.000 randomly drawn model parameter sets, 10 signatures and error metrics are applied individually, in groups and in hierarchical combination to identify behavioral parameter sets from the initial set. The resulting subsets are then discussed with respect to the equifinality reduced (i.e. by how much the initial parameter ranges were narrowed).

The authors show that by jointly applying all criteria considerably narrows the behavioral parameter sets (here: to nine). However, these still show large differences, specifically with respect to catchment groundwater table (values and spatial patterns). From the analysis, the authors conclude that i) a multi-criteria approach to identification of behavioral parameter sets is superior to single-criteria approaches, ii) dynamic constraints to be more effective than non-dynamical ones, and iii) that despite substantial narrowing of the parameter space still large differences among the surviving parameter sets remain, especially with respect to spatial patterns of hydrologic states.

3. Overall ranking

The work is ranked 'Major revision'.

4. Evaluation
Major points
I like the work presented in this paper, especially the strong argument towards using multiple, hard and soft sources of information to identify behavioral model parameter sets, and the thorough literature review. However, despite the fact that the authors' focus in this paper is to present the concept, with the choice of the catchment and time series used, they have clearly missed some very good opportunities to make their results more general and interpretable. More specific:

Judging from the presented time series of discharge and snow water equivalent, the catchments' hydrological function is very simple (one major discharge event during snowmelt, rainfall-runoff events are hardly playing a role). Arguably, a very simple conceptual hydrological model could reproduce this behavior at least as well as the applied model with respect to all discharge-related signatures and metrics, but with much less parameters.

- So why choose this very simple-behaving catchment if the goal is demonstrate the usefulness of a targeted constraining approach for a distributed, physically based model? This way, the model stays well below its potential, and this also means a lot of opportunities for more targeted model parameter evaluation are missed.

We selected this catchment for a number of reasons, but especially because it is a place where modelers and experimentalists have collaborated before, where other types of models have been applied. Most importantly, we selected it because it is a place where we have lots of understanding. We sought to incorporate distributed measurements that are more likely to be available in other catchments (e.g., SWE) to frame our application in a general way that could be applied to other sites. Furthermore, as this is a relatively simple system, it provides a first-order test as to whether this type of approach could have merit before we introduce this approach to a more complex set of catchments.

An additional benefit to this system is it has been modeled previously with conceptual models, as highlighted in the discussion section (page 16, section 5.2). We compare our findings to other simple conceptual models and distributed models of this system (across similar periods), and show that we achieve similar levels of fit. To your point above, we have added a discussion of the choice of model complexity, framed by applications within this specific catchment (page 17):
       "The three models to which we compare our results demonstrate a range of model frameworks that can be used to evaluate model behaviour: conceptual (Smith et al., 2013), lumped (Ahl et al., 2008), and distributed without physically-based parameters (Nippgen et al., 2015). As is shown in this study, all of these models are able to accurately simulate the hydrograph for this catchment. The primary trade-offs across these models include requirements for inputs and parameters alongside computational requirements, which are inversely related to the complexity of simulated behaviour that can be produced from each of these models. While any of these approaches may be used to simulate streamflow, each will enable researchers to answer different questions related to hypotheses about catchment functioning, the use of field information to inform model parameter constraints, and predictions of spatio-temporal

hydrologic processes. Finally, these contrasting models also illustrate the differences between a model like DHSVM that may be applied to many different catchments versus the models introduced by Smith et al. (2013) and Nippgen et al. (2015), in which the model framework and structural equations were developed only for this catchment. In this study, we specifically evaluate the application of physically-based, distributed models to simulate experimental catchments, though we encourage researchers to select the right tool, and therefore the appropriate model, for a given study objective."

- Furthermore, the many degrees of freedom in your model inevitably lead to problems of equifinality, which would not exist in a simpler model appropriate for the simple catchment. So why not choose a more complex catchment, which requires a distributed model?

Selecting a more complex catchment (e.g., in terms of more variable vegetation or soil) would inevitably lead to greater issues with equifinality. Thus, we sought to first constrain this approach using a straightforward framework (e.g., distributed vegetation but undistributed soil types). If the framework were not able to reproduce the hydrograph, this would merit further distribution of inputs. Future work will include evaluation of this type of approach in catchments of differing complexity – e.g., complex inputs, more variable soils, or more variable vegetation. We would suggest any catchment may be modeled using a distributed model if the end goal is to predict distributed hydrologic processes – the simplicity of the catchment does not negate our interest in testing whether observations of streamflow and SWE may constrain the simulation of distributed hydrologic processes.

The use of a distributed model may depend on either the spatial complexity of catchment characteristics, or the desire to simulate spatially-distributed hydrologic behavior. As these types of models are regularly being used to simulate spatially-distributed behavior, regardless of the availability of data to constrain model inputs, parameters, or simulations, we specifically chose a more simplistic catchment where we could maintain a parsimonious number of model parameters.

Along the same lines: Why was a catchment selected with such little available observations as 'hard truth'? Why not choose one with a network of observed groundwater tables, ET, nested discharge observations, spatially distributed information on soil type and soil depth etc.?

We have chosen a site with nested discharge observations, and have presented these nested results in this manuscript. We do not incorporate comparisons to observed well behavior and ET in part because we are interested in the patterns of this information, not just matching these values to a single point. As we discuss on page 8, internal simulations of catchment behavior may still be incorporated into evaluations of distributed model behavior. As internal catchment measurements are often difficult to come by, we have sought to specifically incorporate an evaluation of model predictions in the absence of 'hard truth', to show that this type of evaluation is possible.

In fact, the model elements are all set to the same soil depth and soil type, which makes it much less distributed as it could be.

While we do not distribute soil type or depth, this is in part framed by experimental observations at this site (as outlined on page 5), as well as the strong tradeoff in equifinality that exists as more and more 'types' are added to a given distributed characteristics, contrasted with whether there are enough available observations to truly distribute soil types in space as well as with depth. Thus, we opted to create a parsimonious distributed representation of the system. Vegetation types are distributed based on vegetation height, as we expect this to be an important determinant of system behavior.

Placing the study in a better equipped catchment would offer the opportunity to fill the very nice framework with many more signatures and metrics, especially those evaluating spatial patterns. Furthermore, this would have opened the opportunity to compare the value of constraints formulated as aggregated/heuristic expert knowledge to 'hard' constraints based on observations. This is a clear miss.

From this comment, we have clarified one of our objectives for this study – in the absence of spatially distributed measurements, can point observations, especially streamflow, inform catchment patterns of hydrologic behavior? We have altered the text on page 4 to reflect this point:

"Secondarily, we also explore whether this type of approach, using observations of a subset of hydrologic processes, may inform simulations of other unmeasured spatially-distributed hydrologic processes. Thus, we seek to test whether temporal observations may contain information regarding simulation of hydrologic patterns."

This point is further clarified in the results (4.8) and discussion (5.5) sections.

Along the same lines: All evaluations are done for a single year, and for the calibration period. This way it is impossible to judge
− to which degree the remaining behavioral parameter sets are dependent on the chosen calibration period, and
− whether the behavioral parameter sets found in calibration are still behavioral during a different, validation time. This is a clear miss.

Model predictions are performed for two years – a one year calibration period and a one year validation period (section 2.2.3). We have clarified this point in the text, as it was a source of confusion.

We additionally include all streamflow performance metrics in Table B1 of the Appendix, to enable comparison of results to another period. As can be seen from Table B1, most sets maintain high performance for WY 2007 with respect to streamflow at LSC, with the exception of slightly higher RRE values for sets 3 and 5 and SFDCE for set 4.

While the period considered for model performance is relatively short, it still represents a large computational burden in terms of the size of the catchment being resolved (22.5 km$^2$, 10 m by 10 m resolution) as well as the sampling procedure employed. We expand on this point, as well as

possible ways to address this challenge of computational burden in the future, on page 17, with the following text:

"Ultimately, our ability to resolve issues with equifinality and identify appropriate parameter sets in space and time is challenged, as it was in this study, by the computational demand of complex models. Executing model predictions for the relatively short period of time investigated in this study across 10,000 parameter samples required thousands of computing hours (and even longer periods if the modeller retains or "saves" spatial predictions across the catchment). While distributed, physically-based models like DHSVM may have the ability to resolve predictions of hydrologic processes through space and time, we do not yet have effective, computationally inexpensive approaches for evaluating and representing uncertainties in these types of applications. In order to put these types of models to the test, we need better parameter sampling strategies (e.g., Rakovec et al., 2014; Jefferson et al., 2015) and alternative approaches to those we use for conceptual models, where executing a model many times is not a challenge or limit on analysis. This may come in the form of new methods, or alternatively, approaches that evaluate model adequacy via frameworks for computationally frugal analysis (Hill et al., 2015). While quantifying or limiting equifinality may always be a challenge for physically-based, distributed catchment models, we likely will need to reframe our approaches for evaluating the uncertainties associated with complex model applications. This challenge may be best addressed by encouraging interaction across the conceptual modelling community and the fully, distributed, physically-based modelling community, to address broad issues related to uncertainty and equifinality that, it can be argued, plague all models of any complexity (Hrachowitz and Clark, 2017)."

The authors advocate a multi-criteria approach to identify behavioral parameter sets. However, looking at the criteria in Table 2, two questions arise

Parameterization of ET plays an important role in the model (parameters 22 to 53 in Table 1). However, ET is only used as a very weak constraint (300-650 mm/a). Why not also evaluate the model with respect to ET error, ET timing, ET peaks etc.? From the text, my understanding is that ET estimates from local observations exist. This would be another important and independent criterion.

[Even if the constraints are not very narrow,

We compare our results primarily to streamflow and SWE because we sought to demonstrate results for observations that are likely to be present in other experimental catchments (now clarified on page 10). Additionally, as this is an arid, snowmelt driven system, we chose to compare model simulations to SWE, as we concluded this may be a more important criterion. In a screening sensitivity analysis of model sub-catchments, Kelleher et al. (2015) showed for this system that vegetation parameters (22 to 53 in Table 1) were key to the prediction of metrics assessing SWE. A smaller subset of model parameters directly influenced metrics related to ET. Thus, we conclude that these model parameters are more important determinants of SWE, though are still important to ET, and therefore chose to evaluate model simulations with respect to SWE.

While it may appear that ET is only weakly constrained, we found that, in fact, it is particularly discerning. To assess the impact of ET and other model constraints on the final set of parameters, we calculated the effect of removing a single constraint on the remaining number of parameter sets that meet all other criteria. Out of all possible metric constraints, removing constraint on ET has the biggest impact on the number of final parameter sets. This illustrates that even our weak constraint can have an impact on matching catchment-wide behavior.

[Figure]

**Figure B3: The impact of removing a single constraint on the number of final behavioural parameter sets.**

I do not understand the usage of AI: From my understanding of the text, both PE and P are from observations, so AI is independent of the model. So how can this criterion be used as a signature for model evaluation?

PE is calculated using a Penman-Monteith approach, and therefore is also impacted by vegetation parameters. This is stated on Page 9 lines 23-24.

Minor points
From my experience, the main control on parameter equifinality is model structural choice. The authors discuss this important issue briefly in the conclusions. I encourage them to discuss this aspect in more detail, although I am well aware that model structural choice is not the topic of the paper. However, it can offer an avenue of progress to reduce the still-high equifinality of the final behavioral set of parameter sets.

We have centralized this discussion in section 5.6 with the following text:

"In this vein, the choice of model structure may also offer another opportunity to reduce equifinality (Clark et al., 2008; Pokhrel et al., 2008; Samaniego et al., 2010; Rakovec et al., 2016). In particular, the extensive body of literature on parameter regularization may offer a pathway for maintaining spatial complexity and consistency while reducing the number of free model parameters (Hundecha and Bardossy, 2004; Hundecha et al., 2008; Samaniego et al., 2010; Rakovec et al., 2016). Alternatively, there is also a body of work that treats the model framework itself as a form of uncertainty, testing different model structures as hypotheses for how a catchment may function (Clark et al., 2008; Clark et al., 2011; Fenicia et al., 2011;

Hrachowitz et al., 2014). This approach may also provide an alternative to predicting hydrology via a model with fewer parameters than the distributed application shown here, with a model structure that incorporates the level of detail mandated by the complexities of the catchment (e.g., Euser et al., 2015; Zehe et al., 2014). As encouraged by Beven (2002), to best represent catchment behaviour, we may need to not only focus on model parameters, but also the model structure in terms of how this reflects the physical landscape."

P8/l29: RR instead of PET?
PET varies with model parameters in DHSVM as it is calculated following a Penman-Monteith formulation.

P12/L17: Where is Appendix D?
We have corrected this to 'Table B1'.

P13/L3: Why not also compare simulations to well observations? (see also my major comments)
Please see our response above.

P13/L7: Appendix B3 instead of B only?
We have changed this.

P14/L22-23: … suggest that evaluating certain types of internal behavior by point observations…?
We have corrected this sentence to:
"Together, these results broadly suggest that not all observations will reduce equifinality."

P21/L14: Figure B3 instead of Fig 9?
We leave the reference to Figure 9, but include 'Figure B3' in the previous sentence.

Fig 1: − A) The location of the label 'Tenderfoot creek' is misleading − A) is the scale really [km]? − C) add a legend (which gauge is which)
We have corrected these points in Figure 1.

Fig 4, caption: remaining instead of removed?
We have changed this to 'retained'.

Fig 6, caption: Where are the black dotted lines? Where is (a) and (b)?
We have corrected this figure caption.

Fig 7: Please add year indicators (2007, 2008), a legend (which gauge is which) and for clarity add in the caption that these are plots for the final subset of 9 parameter sets
We have made these changes.

Fig 8, caption: For clarity please add in the caption that these are plots for the final subset of 9 parameter sets
We have altered the caption.

Yours sincerely, Uwe Ehret

References:
Ahl, R. S., Woods, S. W., and Zuuring, H. R.: Hydrologic calibration and validation of swat in a snow-dominated rocky mountain watershed, Montana, USA, J. Am. Water Resour. As., 44, 1411-1430, doi: 10.1111/j.1752-1688.2008.00233.x, 2008.

Beven, K. J.: Towards an alternative blueprint for a physically based digitally simulated hydrologic response modelling system, Hydrol. Proces., 16, 189-206, doi: 10.1002/hyp.343, 2002.

Clark, M. P.,  Slater, A. G., Rupp, D. E., Woods, R. A., Vrugt, J. A., Gupta, H. V., Wagener, T., and  Hay, L. E.: Framework for Understanding Structural Errors (FUSE): A modular framework to diagnose differences between hydrological models, Water Resour. Res., 44, W00B02, doi:10.1029/2007WR006735, 2008.

Clark, M. P., Kavetski, D., and Fenicia, F.: Pursuing the method of multiple working hypotheses for hydrological modelling, Water Resour. Res., 47, W09301, doi:10.1029/2010WR009827, 2011.

Euser, T. Hrachowitz, M., Winsemius, H.C., and Savenije, H. H.: The effect of forcing and landscape distribution on performance and consistency of model structures, Hydrol. Process., 29, 3727-3743, doi: 10.1002/hyp.10445, 2015.

Fenicia, F., Kavetski, D., and Savenije, H. H. G.: Elements of a flexible approach for conceptual hydrological modelling: 1. Motivation and theoretical development, Water Resour. Res., 47, W11510, doi:10.1029/2010WR010174, 2011.

Hill, M. C., Kavetski, D., Clark, M., Ye, M., Arabi, M., Lu, D., Foglia, L. and Mehl, S.: Practical Use of Computationally Frugal Model Analysis Methods, Groundwater, 54, 159–170, doi:10.1111/gwat.12330, 2016.

Hrachowitz, M., Fovet, O., Ruiz, L., Euser, T., Gharari, S., Nijzink, R., Freer, J., Savenije, H. H. G., and C. Gascuel-Odoux: Process consistency in models: The importance of system signatures, expert knowledge, and process complexity, Water Resour. Res., 50, 7445–7469, doi:10.1002/2014WR015484, 2014.

Hrachowitz, M. and Clark, M.: HESS Opinions: The complementary merits of top-down and bottom-up modelling philosophies in hydrology, Hydrol. Earth Syst. Sci. Discuss., doi:10.5194/hess-2017-36, in review, 2017.

Hundecha, Y., and Bardossy, A.: Modeling effect of land use changes on runoff generation of a river basin through parameter regionalization of a watershed model, J. Hydrol., 292, 281–295, doi: 10.1016/j.jhydrol.2004.01.002, 2004.

Hundecha, Y., Ouarda, T. B. M. J., and Bardossy, A.: Regional estimation of parameters of a rainfall-runoff model at ungauged watersheds using the spatial structures of the parameters within a canonical physiographic-climatic space, Water Resour. Res., 44, W01427, doi:10.1029/2006WR005439, 2008.

Jefferson, J. L., Gilbert, J. M., Constantine, P. G., and Maxwell, R. M.: Active subspaces for sensitivity analysis and dimension reduction of an integrated hydrologic model, Comput. Geosci., 90, 78-89, doi:10.1016/j.cageo.2015.11.002, 2016.

Kelleher, C., Wagener, T., and McGlynn, B.: Model-based analysis of the influence of catchment properties on hydrologic partitioning across five mountain headwater subcatchments, Water Resour. Res., 51, 4109–4136, doi:10.1002/2014WR016147, 2015.

Nippgen, F., McGlynn, B. L., and Emanuel, R. E.: The spatial and temporal evolution of contributing areas, Water Resour. Res., 51, 4550– 4573, doi:10.1002/2014WR016719, 2015.

Pokhrel, P., Gupta, H. V., and Wagener, T.: A spatial regularization approach to parameter estimation for a distributed watershed model, Water Resour. Res., 44, W12419, doi:10.1029/2007WR006615, 2008.

Rakovec, O., Hill, M. C., Clark, M. P., Weerts, A. H., Teuling, A. J., and Uijlenhoet, R.: Distributed evaluation of local sensitivity analysis (DELSA), with application to hydrologic models, Water Resour. Res., 50, 409-426, doi: 10.1002/2013WR01463, 2014.

Rakovec, O., Kumar, R., Attinger, S., and Samaniego, L.: Improving the realism of hydrologic model functioning through multivariate parameter estimation, Water Resour. Res., 52, 7779-7792, doi: 10.1002/2016WR019430, 2016.

Samaniego, L., Kumar. R., and Attinger, S.: Multiscale parameter regionalization of a grid-based hydrologic model at the mesoscale, Water Resour. Res., 46, W05523, doi:10.1029/2008WR007327, 2010.

Smith, T., Marshall, L., McGlynn, B., and Jencso, K.: Using field data to inform and evaluate a new model of catchment hydrologic connectivity, Water Resour. Res., 49, 6834–6846, doi:10.1002/wrcr.20546, 2013.

Zehe, E., Ehret, U., Pfister, L., Blume, T., Schröder, B., Westhoff, M., Jackisch, C., Schymanski, S. J., Weiler, M., Schulz, K., Allroggen, N., Tronicke, J., van Schaik, L., Dietrich, P., Scherer, U., Eccard, J., Wulfmeyer, V., and Kleidon, A.: HESS Opinions: From response units to functional units: a thermodynamic reinterpretation of the HRU concept to link spatial organization and functioning of intermediate scale catchments, Hydrol. Earth Syst. Sci., 18, 4635-4655, doi:10.5194/hess-18-4635-2014, 2014.